# Tropospheric NO$_2$ Measurements Using a Three-wavelength Optical Parametric Oscillator Differential Absorption Lidar

Jia Su[1], M. Patrick McCormick[1,*], Matthew S. Johnson[2], John T. Sullivan[3], Michael J. Newchurch[4], Timothy A.Berkoff[5], Shi Kuang[4], Guillaume P. Gronoff[5, 6]

[1]Center for Atmospheric Sciences, Department of Atmospheric and Planetary Sciences, Hampton University, Hampton, Virginia 23668, USA

[2] Earth Science Division, NASA Ames Research Center, Moffett Field, CA, USA

[3]NASA Goddard Space Flight Center, Chemistry and Dynamics Laboratory, Greenbelt, MD 20771, USA

[4]Atmospheric and Earth Science Department, University of Alabama in Huntsville, Huntsville, Alabama, USA

[5]NASA Langley Research Center, Hampton, VA, 23681, USA

[6]Science Systems and Applications, Inc, VA, 23681, USA

*Correspondence to*: M. Patrick McCormick (PAT.MCCORMICK@HAMPTONU.EDU)

**Abstract**

The conventional two-wavelength Differential Absorption Lidar (DIAL) has measured air pollutants such as nitrogen dioxide (NO$_2$). However, high concentrations of aerosol within the planetary boundary layer (PBL) can cause significant retrieval errors using only a two-wavelength DIAL technique to measure NO$_2$. We proposed a new technique to obtain more accurate measurements of NO$_2$ using a three-wavelength DIAL technique based on an Optical Parametric Oscillator (OPO) laser. This study derives the three-wavelength DIAL retrieval equations necessary to retrieve vertical profiles of NO$_2$ in the troposphere. Additionally, two rules to obtain the optimum choice of the three wavelengths applied in the retrieval are designed to help increase the differences of the NO$_2$ absorption cross sections and reduce aerosol interference. NO$_2$ retrieval relative uncertainties caused by aerosol extinction, molecular

extinction, absorption of gases other than the gas of interest and backscattering are calculated
using two-wavelength DIAL (438 nm and 439.5 nm) and three-wavelength DIAL (438 nm,
439.5 nm and 441 nm) techniques.  The retrieval uncertainties of aerosol extinction using the
three-wavelength DIAL technique are reduced to less than 2% of using the two-wavelength
DIAL technique. Moreover, the retrieval uncertainty analysis indicates that the three-wavelength
DIAL technique can reduce more fluctuation caused by aerosol backscattering than two-
wavelength DIAL technique. This study presents $NO_2$ concentration profiles which were
obtained using the HU (Hampton University) three-wavelength OPO DIAL. As a first step to
assess the accuracy of the HU lidar $NO_2$ profiles, we compared the $NO_2$ profiles to simulated
data from WRF-Chem model. This comparison suggests that the $NO_2$ profiles retrieved with the
three-wavelength DIAL technique have similar vertical structure, and magnitudes typically
within ±0.1 ppb, of modeled profiles.
**1. Introduction**
Nitrogen dioxide ($NO_2$) plays a critical role in the tropospheric chemistry and is one of reactive
gases collectively referred to as "nitrogen oxides" ($NO_x$ = nitric oxide and nitrogen dioxide (NO
+ $NO_2$)) [U.S. EPA, 2018]. The sources of $NO_x$ emissions include transportation (on-road
vehicles, airplanes, trains, ships), wood burning, industrial and chemical processes, activities for
oil and gas development, soil emissions, lightning and wildfires (see Nitrogen Oxides Emissions
indicator) [U.S. EPA, 2018]. Once emitted, NO reacts rapidly in the presence of ozone to form
$NO_2$. In U.S. urban locations, most measured airborne $NO_2$ comes from the reaction of these two
precursors, rather than from direct $NO_2$ emissions [Bertram, et al., 2005; Beirle, et al., 2011].
Scientific evidence indicates that short-term $NO_2$ exposure, ranging from 30 minutes to 24 hours,
can cause the exacerbation of asthma symptoms, in some cases resulting in hospitalization
[Berglund, et al., 1993]. Long-term $NO_2$ exposure is likely to have a causal relationship with
respiratory effects, based on evidence for the development of asthma [U.S. EPA, 2016]. And
$NO_2$ will be included in future cycles of the Global Burden of Disease as global exposure
estimates and evidence on their role as independent risk factors accumulates [Larkin et al., 2017].
Additionally, atmospheric processing of $NO_2$ leads to the formation of nitrogen-bearing particles
that can eventually deposit to the surface, causing acidification, nitrogen enrichment, and other
ecological effects [Russell et al., 2012]. Local or global $NO_2$ monitoring is essential for
understanding atmospheric chemistry as well as for human-health and environmental
management and control.
Measurements of the intensity of ultraviolet or visible absorption spectra from the ground or
from satellites are commonly used to retrieve the column density of $NO_2$ [Celarier et al., 2008;
Valks et al., 2011; Berg et al., 2012]. Satellite-based instruments such as Ozone Monitoring
Instrument (OMI), Global Ozone Monitoring Experiment (GOME and GOME-2) and SCanning
Imaging Absorption SpectroMeter for Atmospheric CHartographY (SCIAMACHY) can provide
global scale $NO_2$ column measurements during daytime [Boersma et al., 2008; Bucsela et al.,
2008]. Moreover, plumes of $NO_2$ by cities, power plants, and even ships can be tracked using the
recent high spatial resolution observations of $NO_2$ from TROPOMI on Sentinel-5P since 2017
[Lorente, et al., 2019; Georgoulias et al., 2020]. However, they are unable to obtain local high
temporal resolution $NO_2$ emissions such as variations in hourly $NO_2$ concentrations due to their
long repeat cycle, since the lifetime of tropospheric $NO_2$ is only about 6 hour in summer and 18-
24 hours in winter due to photochemical effect [Beirle, et al., 2003; Cui et al., 2016]. In addition,
measurements of tropospheric $NO_2$ from satellite or aircraft are also influenced and limited by
clouds [Bovensmann et al., 1999; Liang et al., 2017]. Ground-based measurements of column
NO$_2$ from instruments such as Pandora using differential optical absorption spectroscopy (DOAS)
are often used for the validation of satellite instruments [Herman et al., 2009; Lamsal et al., 2014;
Kollonige et al., 2018]. In situ measurements of near-surface NO$_2$ can best monitor local
emissions. However, at this point in time, they cannot provide vertically-resolved measurements.
Balloon measurements using a NO$_2$-sonde can produce vertical profiles, but these measurements
are very limited in time and space, especially in the Southern Hemisphere. The primary source of
data on the vertical distribution of NO$_2$ comes from operational sites around the world. However,
their operation can be expensive and labor-intensive. [Scott et al., 1999; Herman et al., 2009;
Sluis et al., 2010].
The DIAL technique offers the potential for autonomous, 24x7 operation, with improved
temporal resolution. Absorption of light by molecules is the basis for DIAL and numerous
atmospheric constituents absorbing light. Conventional DIAL operates at two absorbing
wavelengths, one stronger than the other indicated by on ($\lambda_{on}$) and off ($\lambda_{off}$) wavelength of the
gaseous absorption feature of interest. Because of different absorption at $\lambda_{on}$ and $\lambda_{off}$, the
difference between the backscattered laser signals at the two wavelengths can be used to derive
the number density of the absorption gas. Taking the log-ratio of these returns at closely spaced
wavelengths removes system parameters and attenuation to and from the target of interest [Rothe
et al., 1974; Sullivan et al., 2014]. Thus, this technology provides measurements of the
concentration of gas, such as NO$_2$, O$_3$, and SO$_2$ at a particular location and time [Fredriksson et
al., 1984; Newchurch et al., 2003; Kuang et al., 2013; Sullivan et al., 2017]. The DIAL technique
provides the unique capability of remotely monitoring urban/rural area localized NO$_2$
concentrations/emissions and profiling their tropospheric vertical NO$_2$ concentration. However,
aerosols are abundant within the PBL and can cause significant retrieval errors in a two-
wavelength DIAL technique to measure $NO_2$. To better understand this aerosol problem and
produce a more accurate $NO_2$ profile measurement, we described a new technique using a three-
wavelength DIAL technique based on the intrinsic capabilities of using a multi-wavelength OPO
laser system. HU has incorporated an OPO laser into its lidar system. The OPO laser enables
researchers to optimize (tune) wavelength choices for specific measurements [P.Weibring et al.,
2003]. The three-wavelength DIAL retrieval equations are derived in this study.  Our optimum
choices for the three wavelengths to be used for our $NO_2$ retrievals are designed to help increase
the difference in $NO_2$ absorption cross section, and reduce aerosol influence.  $NO_2$ retrieval
relative uncertainties are calculated using the two-wavelength DIAL (438 nm and 439.5 nm) and
the three-wavelength DIAL (438 nm, 439.5 nm and 441 nm). Tropospheric $NO_2$ profiles were
obtained by applying the proposed technique to HU OPO DIAL lidar. As a first-order assessment,
the HU lidar results were compared with simulated data from the WRF-Chem air quality model.
**2. Method**
To minimize aerosols-interference on the retrievals of $NO_2$, a three-wavelength DIAL technique
was proposed with $\lambda_1 < \lambda_2 < \lambda_3$. Table 1 shows expressions for the extinction and backscatter of
molecules and aerosols for these three wavelengths. In Table 1, $\beta_m$ and $\beta_a$ are backscatter from
molecules and aerosols for the wavelength of $\lambda_2$; $\alpha_m$ and $\alpha_a$ are the extinction of molecules and
aerosols for the wavelength of $\lambda_2$; $e$ is the aerosol Ångström exponent and assumed to be equal
for the three wavelengths because the three wavelengths are very close.

Table 1. Extinction and backscatter of molecule and aerosol for wavelengths of $\lambda_1$, $\lambda_2$ and $\lambda_3$.

| wavelength | Molecular backscattering | Aerosol backscattering | Molecular extinction | Aerosol extinction |
|---|---|---|---|---|
| $\lambda_1$ | $\left(\frac{\lambda_1}{\lambda_2}\right)^{-4} \beta_m$ | $\left(\frac{\lambda_1}{\lambda_2}\right)^{-e} \beta_a$ | $\left(\frac{\lambda_1}{\lambda_2}\right)^{-4} \alpha_m$ | $\left(\frac{\lambda_1}{\lambda_2}\right)^{-e} \alpha_a$ |
| $\lambda_2$ | $\beta_m$ | $\beta_a$ | $\alpha_m$ | $\alpha_a$ |

| | | | | |
|---|---|---|---|---|
| $\lambda_3$ | $\left(\frac{\lambda_3}{\lambda_2}\right)^{-4}\beta_m$ | $\left(\frac{\lambda_3}{\lambda_2}\right)^{-e}\beta_a$ | $\left(\frac{\lambda_3}{\lambda_2}\right)^{-4}\alpha_m$ | $\left(\frac{\lambda_3}{\lambda_2}\right)^{-e}\alpha_a$ |

The three elastic lidar equations can be expressed as:

$$X(\lambda_1,Z) = C_1 \frac{\left[\left(\frac{\lambda_1}{\lambda_2}\right)^{-4}\beta_m(Z)+\left(\frac{\lambda_2}{\lambda_1}\right)^{-e}\beta_a(Z)\right]}{Z^2}\exp\left\{-2\int_0^Z\left[\left(\frac{\lambda_1}{\lambda_2}\right)^{-4}\alpha_m(z)+\left(\frac{\lambda_2}{\lambda_1}\right)^{-e}\alpha_a(z)+\sigma_N(\lambda_1,z)N_N(z)+O_{abs}(\lambda_1,z)\right]dz\right\} \quad (1)$$

$$X(\lambda_2,Z) = C_2 \frac{[\beta_m(Z)+\beta_a(Z)]}{Z^2}\exp\{-2\int_0^Z[\alpha_m(z)+\alpha_a(z)+\sigma_N(\lambda_2,z)N_N(z)+O_{abs}(\lambda_2,z)]dz\} \quad (2)$$

$$X(\lambda_3,Z) = C_3 \frac{\left[\left(\frac{\lambda_3}{\lambda_2}\right)^{-4}\beta_m(Z)+\left(\frac{\lambda_3}{\lambda_2}\right)^{-e}\beta_a(Z)\right]}{Z^2}\exp\{-2\int_0^Z[\left(\frac{\lambda_3}{\lambda_2}\right)^{-4}\alpha_m(z)+\left(\frac{\lambda_3}{\lambda_2}\right)^{-e}\alpha_a(z)+\sigma_N(\lambda_3,z)N_N(z)+O_{abs}(\lambda_3,z)dz\} \quad (3)$$

where X is the lidar signal; $C_1$, $C_2$ and $C_3$ are lidar constants; the subscripts $a$ and $m$ represent aerosol, and molecule, respectively; $\sigma_N$ is the absorption cross section for the gas of interest; $N_N$ is the molecular density of the gas of interest; $O_{abs}$ is absorption of gases other than the gas of interest and $z$ is the altitude. The molecular density of the gas of interest can be obtained using Eq. (1), (2) and (3).

NO$_2$ density retrieval equation can be expressed as:

$$N_N(Z) = \frac{\frac{1}{2}\times\frac{d}{dz}\left[ln\frac{X(\lambda_1,Z)X(\lambda_3,Z)}{X(\lambda_2,Z)^2}\right]-AED(z)-MED(z)-OAD(z)-B(z)}{\Delta\sigma_N} \quad (4)$$

$$\Delta\sigma_N = 2\sigma_N(\lambda_2)-\sigma_N(\lambda_1)-\sigma_N(\lambda_3) \quad (5)$$

$$B(z) = \frac{1}{2}\frac{d}{dz}\left[ln\frac{\left[\left(\frac{\lambda_3}{\lambda_2}\right)^{-4}\beta_m(Z)+\left(\frac{\lambda_3}{\lambda_2}\right)^{-e}\beta_a(Z)\right]\left[\left(\frac{\lambda_1}{\lambda_2}\right)^{-4}\beta_m(Z)+\left(\frac{\lambda_1}{\lambda_2}\right)^{-e}\beta_a(Z)\right]}{[\beta_m(Z)+\beta_a(Z)]^2}\right] \quad (6)$$

$$AED(z) = K\alpha_a(Z) \qquad K = 2-\left(\frac{\lambda_1}{\lambda_2}\right)^{-e}-\left(\frac{\lambda_3}{\lambda_2}\right)^{-e} \quad (7)$$

$$MED(z) = \left[2-\left(\frac{\lambda_1}{\lambda_2}\right)^{-4}-\left(\frac{\lambda_3}{\lambda_2}\right)^{-4}\right]\alpha_m(Z) \quad (8)$$

$$OAD(z) = 2O_{abs}(\lambda_2,z)-O_{abs}(\lambda_1,z)-O_{abs}(\lambda_3,z) \quad (9)$$

where AED, MED, OAD and B are the correction terms of aerosol extinction, molecular extinction, absorption of gases other than the gas of interest and backscattering, respectively. Because the atmospheric molecular density is relatively stable, MED can be corrected using a numerical model or local real-time radiosonde data. OAD can be removed by choosing

appropriate wavelengths. However, aerosol is variable especially in PBL. For correction of AED
and B, we need accurate aerosol measurements. However, accurate aerosol measurements are not
easily to be obtained. From the above $NO_2$ retrieval relative equation, all of correction terms are
related to the three wavelengths, so how to choose the three wavelengths is very critical to
reduce correction terms and improve the accuracy of $NO_2$ retrievals. We designed two rules to
obtain the optimum choice for the three wavelengths:
**a.** The chosen three wavelengths increase differences of the $NO_2$ absorption cross section $(\Delta\sigma_N)$
to improve $NO_2$ retrieval.
According to Eq. (4), the more $\Delta\sigma_N$ is, the less all of correction terms are. So the chosen three
wavelengths should help to increase $\Delta\sigma_N$. Generally, researchers only used an increasing
absorption method $(\sigma_N(\lambda_1)<\sigma_N(\lambda_2)<\sigma_N(\lambda_3))$ or a decreasing absorption method $(\sigma_N(\lambda_1)>$
$\sigma_N(\lambda_2)>\sigma_N(\lambda_3))$ (illustrated in Fig. 1) to choose the three wavelengths [Wang, et al., 1997; Liu, et
al., 2017]. Wang used three wavelengths corresponding to the strong, medium and weak
absorption of $O_3$ to obtain an accurate stratospheric ozone profile in the presence of volcanic
aerosols. Liu used three wavelengths of 448.10nm, 447.20nm and 446.60 nm corresponding to
the strong, medium and weak absorption of $NO_2$ to retrieve $NO_2$. Equation (10) and (11) are
calculated values of $\Delta\sigma_N$ for the increasing absorption method and the decreasing absorption
method using Eq. (5). Using the increasing absorption method and the decreasing absorption
method to choose the three wavelengths, the values of $\Delta\sigma_N$ are both decreased according to Eq.
(10) and (11) compared to the conventional two-wavelength DIAL technique. According to
characteristics of the $NO_2$ absorption spectrum showed in Fig. 2, a bumping absorption method
$(\sigma_N(\lambda_1)<\sigma_N(\lambda_2)\&\sigma_N(\lambda_3)<\sigma_N(\lambda_2))$ is designed to choose the three wavelengths which can
increase value of $\Delta\sigma_N$ compared to the two-wavelength DIAL technique according to Eq. (12).
However, for DIAL systems to measure other atmospheric gases like ozone, it is only practical to
use wavelength selection Method B because of the shape of the ozone absorption spectrum
(lacking narrow peaks).

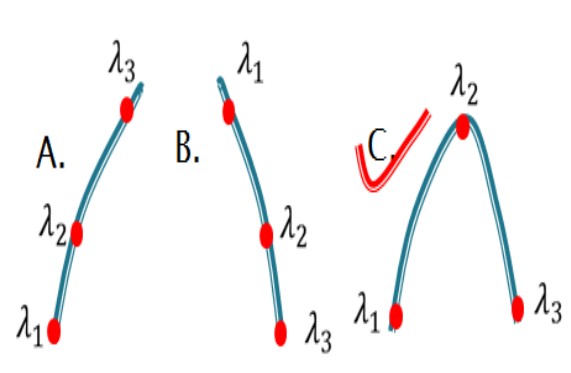

Fig.1 The three-wavelength chosen methods:        Fig.2 $NO_2$ strong absorption cross section
Increasing absorption method (A),  Decreasing      between 420 nm and 450 nm
absorption method (B) and Bumping absorption
method (C)
Increasing absorption method: $\Delta\sigma_N = abs[\sigma_N(\lambda_2) - \sigma_N(\lambda_1)] - abs[\sigma_N(\lambda_2) - \sigma_N(\lambda_3)]$

164                                                                                            (10)

Decreasing absorption method: $\Delta\sigma_N = abs[\sigma_N(\lambda_2) - \sigma_N(\lambda_3)] - abs[\sigma_N(\lambda_2) - \sigma_N(\lambda_1)]$

166                                                                                            (11)

Bumping absorption method: $\Delta\sigma_N = abs[\sigma_N(\lambda_2) - \sigma_N(\lambda_1)] + abs[\sigma_N(\lambda_2) - \sigma_N(\lambda_3)]$

168                                                                                            (12)

**b.** The chosen three wavelengths can reduce or remove AED.
It means the value of AED is equal or close to 0. Choosing the appropriate three wavelengths to
make the value of K in Eq. (12) equal or close to 0, the value of AED will be equal or close to 0.
The value of K in Eq. (12) changes with different aerosol Ångström exponents. For example, to
remove boundary layer aerosol influence, we can set aerosol Ångström exponents=1 to calculate
the value of K to choose the three wavelengths because the size of aerosol in the boundary layer
is typically large [Schuster, et al., 2006].
**3. HU three-wavelength OPO DIAL system**

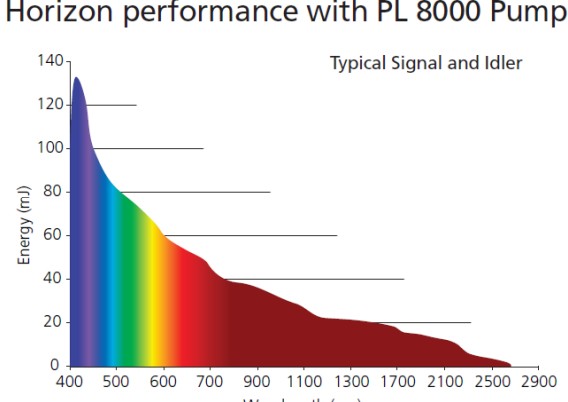

| Description | |
|---|---|
| Pulsewidth (nsec) | 3-7 |
| Pointing Stability (urad) | <±100 |
| Linewidth (cm$^{-1}$) | |
| Unseeded | 3-7 |
| Doubling/Mixing | <10 |
| Energy Stability (%,99% of shots) | <±10 |
| Divergence (mrad,FWHM) | <2 (both axes) |
| Beam Diameter (mm, near field) | 4-7 |
| Beam Roundness (%, near field) | >85 |
| Polarization (%) | |
| Signal Horizontal | >99 |
| Idle Horizontal | >99 |


178        Fig.3 Continuum Horizon II energy outputs (a) and parameters (b) with PL 8000 pump

The HU lidar is located on the campus of HU (37.02° N, 76.34° W) in Hampton, VA. A
Continuum Horizon II tunable OPO laser and a Continuum Powerlite DLS 8000 pump laser have
recently been incorporated into HU lidar system. The OPO laser enables researchers to optimize
(tune) the wavelength choices and provides more flexibility than fixed-frequency wavelength
shifters such as Raman cells. The wavelength tuning range of our OPO extends from 192 nm to
2750 nm. This range is fully automated with precision scanning for true hands-free operation.
Fig. 3(a) and (b) show the Continuum Horizon II output energy and its parameters. The OPO
laser energy outputs between 400 nm and 500 nm which overlap with the $NO_2$ strong absorption
spectral zone in Fig. 2 produce near the maximum possible power in the spectrum. Combining
the OPO laser energy outputs, $NO_2$ absorption spectral and two three-wavelength chosen rules,
438 nm, 439.5 nm and 441 nm shown in Fig. 2 result in the wavelengths of HU three-wavelength
DIAL system because $\Delta\sigma_N$ of the three-wavelength pair is more than other three-wavelength
pairs in $NO_2$ strong absorption spectral zone and the *K* value of the three-wavelength is 0.000023
(close to 0). The HU lidar system currently consists of a Continuum OPO laser system as the
light source, a 48-inch non-coaxial Cassegrainian-configured telescope receiver, a light
separation system that uses beam splitters and interference filters, a detecting system including
photomultiplier tubes (PMT) and avalanche photodiodes (APDs) and a Licel optical transient
recorder. A schematic of the lidar system is shown in Fig.4. The system can be configured to
measure multi-wavelength aerosols and $NO_2$ density. High-resolution backscatter measurements
extend from the boundary layer (1.2 km) to free troposphere. The pump laser operates at three
fixed wavelengths (1064, 532, and 354.7 nm). The 354.7-nm laser is mostly reflected into OPO
laser to produce three-wavelength (338 nm, 339.5 nm and 441 nm). Steering mirrors whose axes
are aligned with a receiving telescope axis directs these laser outputs into the atmosphere. The
laser backscatter is collected by a 48-inch diameter telescope and split into specific wavelength
bands by a beam separation unit, which combines filters and beam-splitters for dispersion of the
return backscatter to various detection channels. Using filters and beam-splitters makes the
beam-splitting system simple, compact, and easy to change or add other spectral channels for
other measurements. Currently, wavelengths of 438 nm, 439.5 nm, 441 nm, 354.7 nm, 532 nm
and 1064 nm are focused to PMTs and APD, and recorded by a Licel data-collecting system for
measurements of aerosol, and $NO_2$.
To demonstrate that the HU three-wavelength OPO DIAL system can effectively reduce aerosol
influence and accurately retrieve $NO_2$, retrieval correction terms of AED, MED, OAD and B in
Eq. (4) are simulated using two-wavelength DIAL technique (438 nm and 439.5 nm)  and the
three-wavelength DIAL technique (438 nm, 439.5 nm and 441 nm). Ozone was used for the
simulation of OAD because only ozone absorption can produce a little influence on $NO_2$ retrieval
based on HITRAN 1.1.2.0 database. Atmospheric data of aerosol, molecule, $O_3$ and $NO_2$ for
these simulations are from the HU local lidar aerosol measurements, radiosonde, NASA
Tropospheric Ozone Lidar Network (TOLNet) and NASA Deriving Information on Surface
Conditions from COlumn and VERtically Resolved Observations Relevant to Air Quality
(DISCOVER-AQ) measurements shown in Fig.5. Extinction and backscatter of aerosol at 438
nm, 439.5 nm and 441 nm can be calculated from aerosol extinction profile at 532 nm in Fig.5 (a)
with the setting of lidar ratio=50 and $e$=1, 2 and 3. Lidar ratio is wavelength dependent and its
value in the visible band is in general smaller than in the UV band for the same type of aerosols
[Kuang et al., 2020; Reid et al., 2017]. Absorption of $NO_2$ and $O_3$ at 438 nm, 439.5 nm and 441
nm can be calculated using their mixing ratio profiles in Fig.5 (b) and their absorption cross-
sections from HITRAN 1.1.2.0 database. MED, AED, OAD, B and absorption difference of $NO_2$
(NAD) are simulated using two-wavelength DIAL technique with different aerosol Ångström
exponents ($e$=1, 2 and 3) shown in Fig. 6 (a), (c) and (e), and the three-wavelength DIAL
technique shown in Fig.6 (b), (d) and (f). In Fig. 6, red lines are NAD; black lines are MED;
deep blue lines are AED; light blue lines are OAD. In Fig. 6, all OAD is far less than NAD. It is
concluded that ozone absorption has negligible influence on the retrieval of $NO_2$. In Fig. 6 (a), (c)
and (d), MED and AED in PBL are both more than NAD using the two-wavelength DIAL
technique. Because atmospheric molecules are relatively stable, MED can be corrected using
local model or real-time radiosonde data. However, aerosol is variable, so aerosols are a
significant uncertainty for retrieving $NO_2$ with the conventional two-wavelength DIAL technique.
In Fig. 6 (b), (d) and (f),  MED and AED in boundary layer are both much smaller than NAD
using proposed three-wavelength DIAL technique. It is proven that three-wavelength DIAL
technique can effectively decrease retrieval errors caused by aerosol extinction. From Fig.5, we
can see AED using three-wavelength DIAL technique can be reduced to less than 2% of AED
using two-wavelength DIAL technique at least. Therefore, even if AED is not corrected, $NO_2$
still can be accurately retrieved. Moreover, simulated B using the two-wavelength DIAL
technique and the three-wavelength DIAL technique are shown in Fig. 6 with green lines. The
sharp change on vertical adjacent aerosol backscatter can cause drastic changes of B term. In Fig.
6, the value of B term using three-wavelength DIAL technique is far less than using two-
wavelength DIAL technique. So the three-wavelength DIAL technique can reduce more
fluctuation caused by aerosol backscattering than two-wavelength DIAL technique.

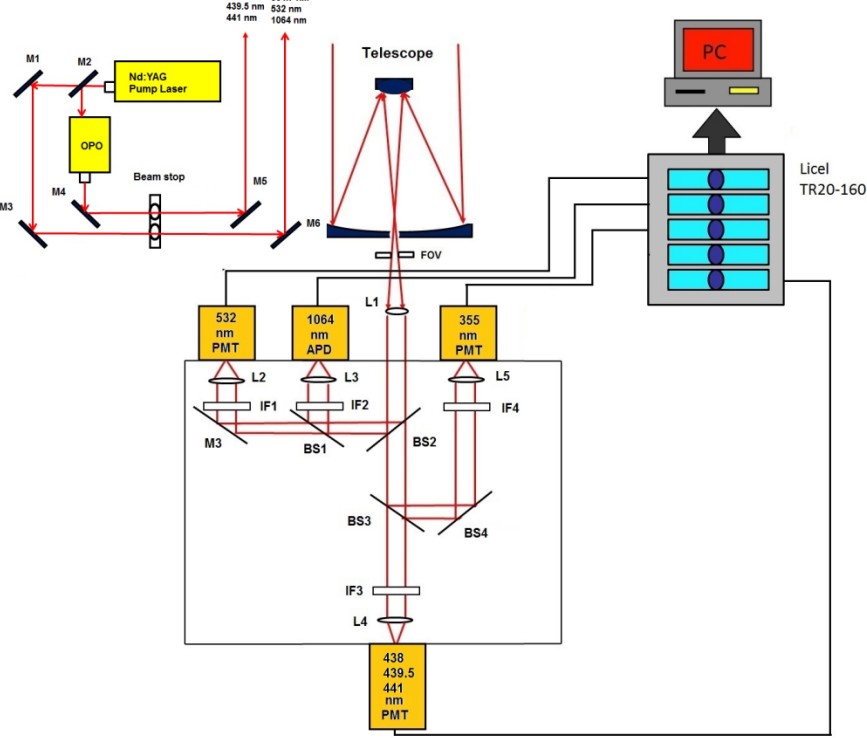


Fig.4 HU lidar system (L-lens, M-mirror, BS-beam-splitter, IF-interference filter, FOV-field of view,
PMT-Photomultiplier tube, APD-Avalanche Photodetector)

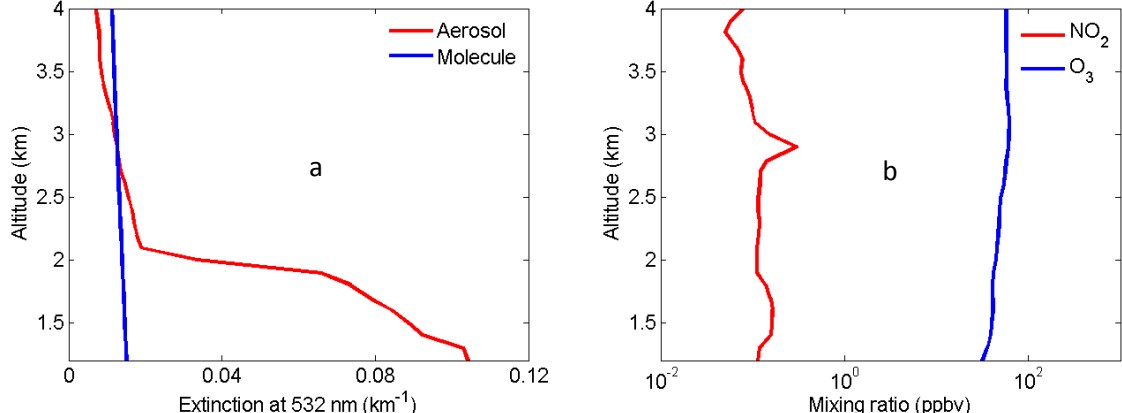


Fig.5 Atmospheric profiles used for modeling $NO_2$ lidar correction terms. (a) Aerosol extinction profile
(red) at 532 nm measured by the HU lidar and molecular extinction profile (blue) at 532 nm derived from
local radiosonde data; (b) $NO_2$ (red) and $O_3$ (blue) mixing ratio profiles from NASA DISCOVER-AQ
and  TOLNet.

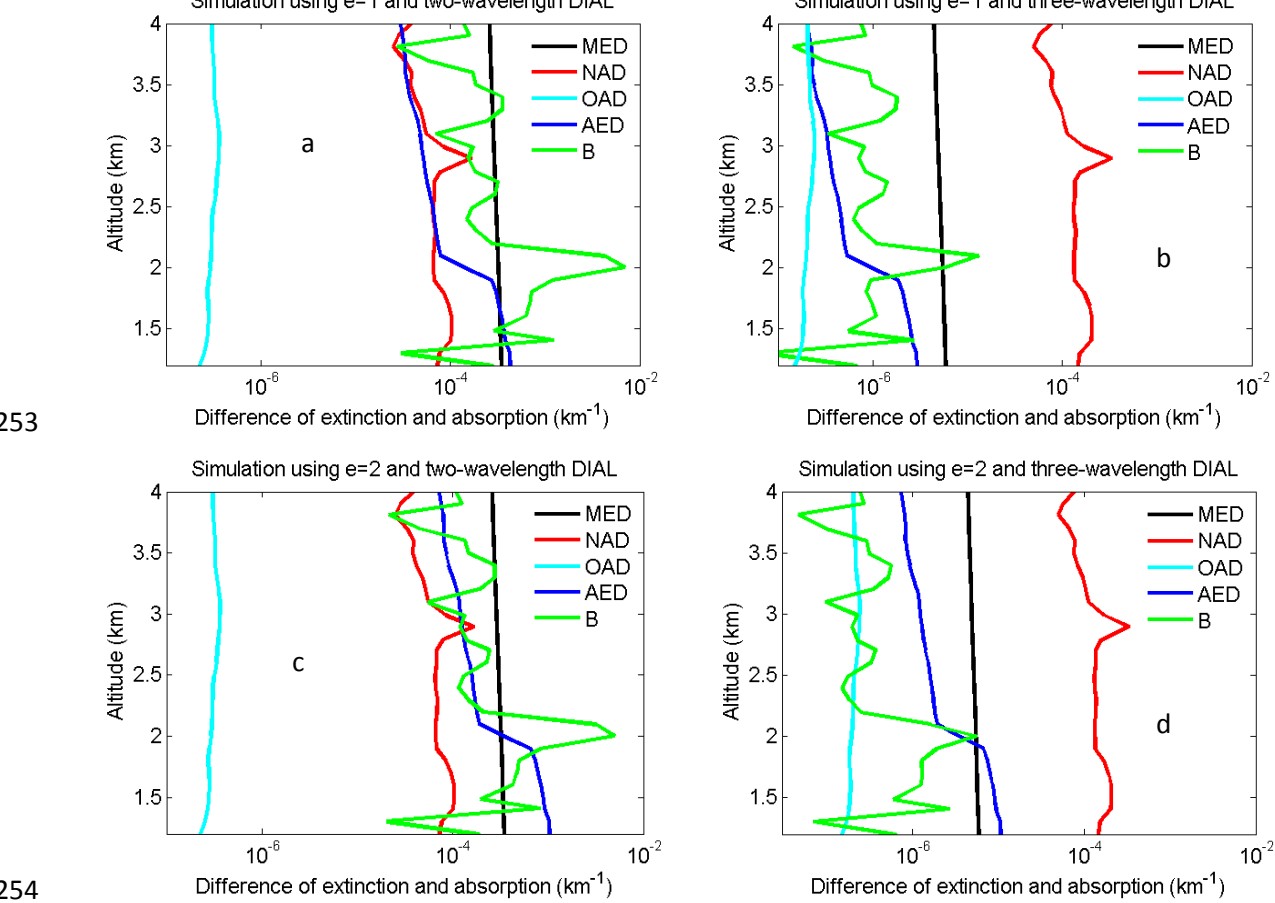



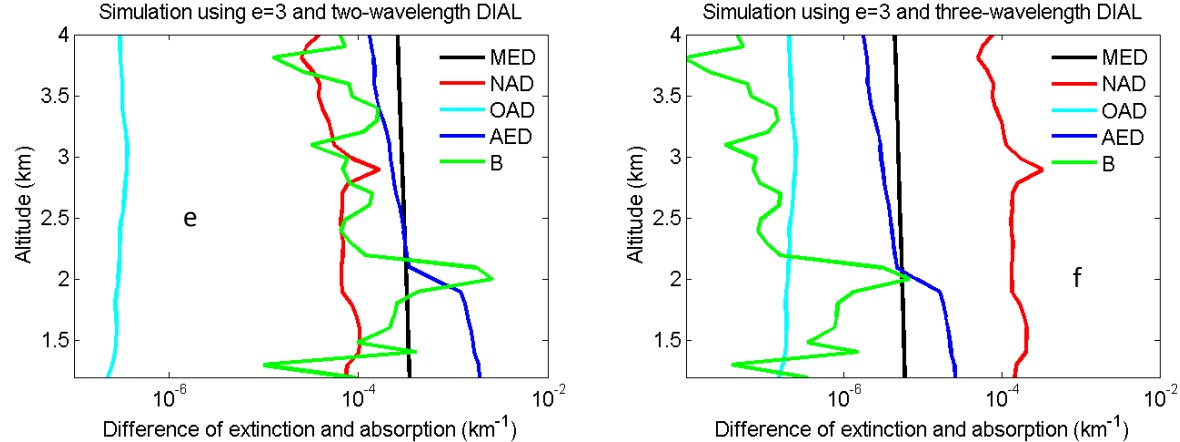

Fig.6 Simulated MED (black), NAD (red), OAD (light blue), AED (deep blue)and B (green) using two-
wavelength DIAL with e=1(a), e=2 (c)  and e=3 (e)  and three-wavelength DIAL technique with e=1(b),
e=2(d) and e=3(f).
**4. Uncertainty analysis**
According to $NO_2$ retrieval Eq.4, the $NO_2$ measurement uncertainty is due to molecule,
absorption of gases other than the gas of interest, aerosol and noise of lidar signals. The total
relative uncertainty can be expressed as Eq. (13) [Leblanc et al., 2016].
$$U_{NO_2}(z) = \sqrt{U_{AED}(z)^2 + U_{MED}(z)^2 + U_{OAD}(z)^2 + U_B(z)^2 + U_S(z)^2}$$ (13)
$$U_{MED}(z) = \frac{u[MED(z)]}{N_N(z)\Delta\sigma_N} = \frac{\left[2-\left(\frac{\lambda_2}{\lambda_1}\right)^4-\left(\frac{\lambda_2}{\lambda_3}\right)^4\right]u[\alpha_m(z)]}{N_N(Z)\Delta\sigma_N} = \frac{\left[2-\left(\frac{\lambda_2}{\lambda_1}\right)^4-\left(\frac{\lambda_2}{\lambda_3}\right)^4\right]\sigma_m u[N_m(z)]}{N_N(Z)\Delta\sigma_N}$$ (14)

$$U_{OAD}(z) = \frac{u[OAD(z)]}{N_N(z)\Delta\sigma_N} = \frac{u[2O_{abs}(\lambda_2,z)-O_{abs}(\lambda_1,z)-O_{abs}(\lambda_3,z)]}{N_N(Z)\Delta\sigma_N} = \frac{[2\sigma_o(\lambda_2,z)-\sigma_o(\lambda_1,z)-\sigma_o(\lambda_3,z)]u[N_o(z)]}{N_N(Z)\Delta\sigma_N}$$ (15)

$$U_{AED}(z) = \frac{u[AED(z)]}{N_N(Z)\Delta\sigma_N} = \frac{\left[2-\left(\frac{\lambda_2}{\lambda_1}\right)^e-\left(\frac{\lambda_2}{\lambda_3}\right)^e\right]u[\alpha_a(z,s)]}{N_N(Z)\Delta\sigma_N}$$ (16)
$$U_B(z) = \frac{u\left\{\frac{1}{2}\frac{d}{dz}\left[ln\frac{\left[\left(\frac{\lambda_2}{\lambda_3}\right)^4\beta_m(z)+\left(\frac{\lambda_2}{\lambda_3}\right)^e\beta_a(z,s)\right]\left[\left(\frac{\lambda_2}{\lambda_1}\right)^4\beta_m(Z)+\left(\frac{\lambda_2}{\lambda_1}\right)^e\beta_a(z,s)\right]}{[\beta_m(Z)+\beta_a(Z,s)]^2}\right]\right\}}{N_N(Z)\Delta\sigma_N}$$ (17)
$$U_S(z) = \frac{u\left\{\frac{1}{2}\times\frac{d}{dz}\left[ln\frac{X(\lambda_1,Z)(\lambda_3,Z)}{X(\lambda_2,Z)^2}\right]\right\}}{N_N(Z)\Delta\sigma_N} =$$
$$\frac{\frac{1}{2}\times\sqrt{\left\{\frac{d\left\{\frac{d}{dz}\left[ln\frac{X(\lambda_1,Z)(\lambda_3,Z)}{X(\lambda_2,Z)^2}\right]\right\}}{d[X(\lambda_1,Z)]}\times u[X(\lambda_1,Z)]\right\}^2 + \left\{\frac{d\left\{\frac{d}{dz}\left[ln\frac{X(\lambda_1,Z)(\lambda_3,Z)}{X(\lambda_2,Z)^2}\right]\right\}}{d[X(\lambda_2,Z)]}\times u[X(\lambda_2,Z)]\right\}^2 + \left\{\frac{d\left\{\frac{d}{dz}\left[ln\frac{X(\lambda_1,Z)(\lambda_3,Z)}{X(\lambda_2,Z)^2}\right]\right\}}{d[X(\lambda_3,Z)]}\times u[X(\lambda_3,Z)]\right\}^2}}{N_N(Z)\Delta\sigma_N}$$

273 (18)

where $U_{NO2}$ is $NO_2$ total retrieval relative uncertainty using three-wavelength DIAL technique;
$U_{MED}$, $U_{OAD}$, $U_{AED}$, $U_B$ and $U_s$ are $NO_2$ retrieval relative uncertainty caused by molecule,
absorption of gases other than the gas of interest, aerosol (extinction and backscattering) and
noise of lidar signals expressed as Eq. (14), (15), (16), (17) and (18); $u$ is uncertainty function;
$N_m$ and $N_o$ are number density (ND) of air and ozone; $\sigma_m$ is Rayleigh scattering cross section; $\sigma_o$
is absorption cross section of ozone; $S$ is lidar ratio.
From Eq. (14) and (15), $U_{MED}$ and $U_{OAD}$ are determined by $u[N_m(z)]$ and $u[N_o(z)]$ (uncertainties
of $N_m$ and $N_o$). In our measurements, profiles of temperature and pressure from local radiosonde
are used to calculate $N_m$. Usually, one radiosonde is launched for about 8-hour measurement.
One profile of air number density from local radiosonde is used to correct 8-hour $NO_2$
measurements. According to statistics of 8-hour variation of temperature and pressure in local
four seasons, the uncertainty of $N_m$ is between 1% and 3%. $U_{MED}$ using two-wavelength DIAL
technique and the three-wavelength DIAL technique are calculated according to Eq. (14) with
the uncertainty of $N_a$ as 1%, 2% and 3% shown in Fig. 7(a). $U_{MED}$ using three-wavelength DIAL
technique is far less than using two-wavelength DIAL technique. $N_o$ is obtained from local
measurements. Because of very low values of ozone absorption cross section differentials, with
the uncertainty of $N_o$ as 50% and 100%, $U_{OAD}$ using two-wavelength DIAL technique and using
the three-wavelength DIAL technique are both less 0.5% from Fig.7 (b). Ozone absorption
correction is neglect in $NO_2$ retrieval. From Eq. (16) and (17), $U_{AED}$ and $U_B$ are determined by
uncertainties of $a_a$, $\beta_a$ and $e$. For HU lidar system, 532-nm elastic signals are used to calculate $a_a$
and $\beta_a$ with Fernald method to correct $NO_2$ retrieval [Fernald et al., 1972]. 50 sr is usually chosen
as lidar ratio to retrieve $a_a$ and $\beta_a$. The lidar ratio is variable, so uncertainties of $a_a$ and $\beta_a$ are
caused by chosen lidar ratio. The range of lidar ratio is about from 30 sr to 70 sr for 532 nm. The
uncertainty of lidar ratio is 40% for 50 sr. The uncertainties of $a_a$ and $\beta_a$ are calculated with
uncertainty of lidar ratio as 40%. Finally, $U_{AED}$ and $U_B$ using two-wavelength DIAL technique
and using the three-wavelength DIAL technique are calculated with the Ångström exponent as 1,
2 and 3 shown in Fig. 8, 9 and 10. From these figures, $U_{AED}$ and $U_B$ using three-wavelength
DIAL technique are both less 4%. However, $U_{AED}$ below 2 km using two-wavelength DIAL
technique are more than 90% after correction of aerosol extinction. From Eq. (18), $U_s$ is
determined by uncertainties of three-wavelength lidar signals. The uncertainties of lidar signals
with average integration time of 1 minute and 2 minutes are derived from Poisson statistics
associated with the probability of detection of a repeated random event [Measures, 1984;
Leblanc et al., 2016]. $NO_2$ number density relative uncertainty owing to the noise of lidar signals
with average integration time of 1 minute and 2 minutes are obtained shown in Fig11. We can
see $U_s$ using two-wavelength DIAL technique is smaller than using three-wavelength DIAL
technique. With increase of average integration time from 1 minute to 2 minutes, $U_s$ can be
effectively reduced. At last, $U_{NO_2}$ (the total relative uncertainties of $NO_2$) with e as 1, 2 and 3 are
calculated shown in Fig. 12(a), (b) and (c).

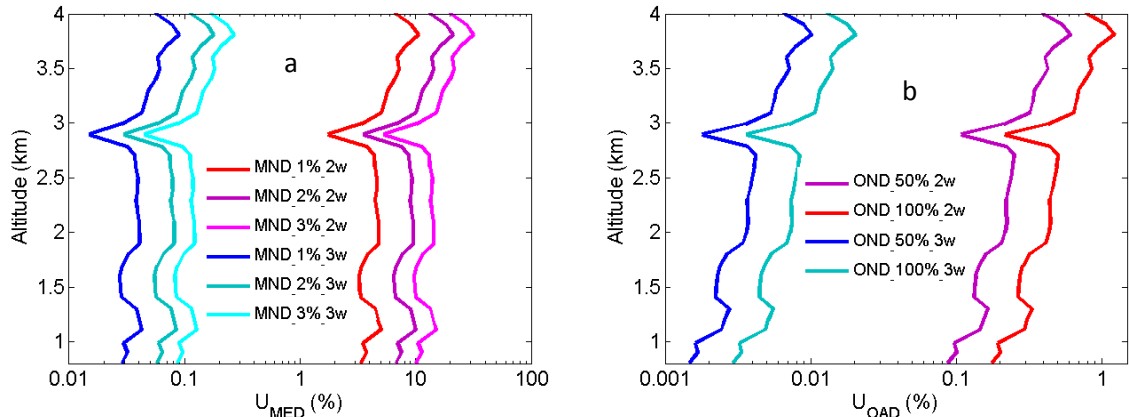


Fig.7 NO$_2$ number density relative uncertainty owing to air number density (a) and ozone number density (b).

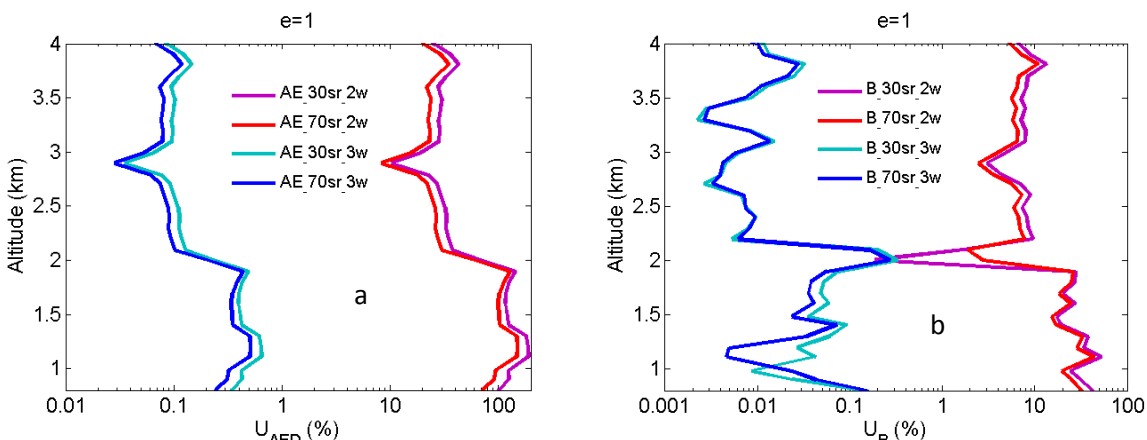


Fig.8 NO$_2$ number density relative uncertainty owing to aerosol extinction (a) and backscatter (b) with e= 1.


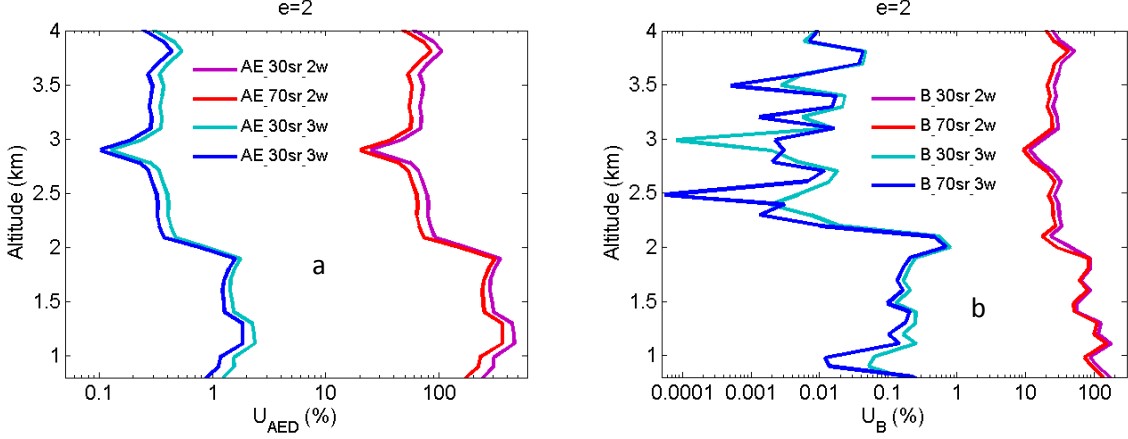


Fig.9 NO$_2$ number density relative uncertainty owing to aerosol extinction (a) and backscatter (b) with e= 2.

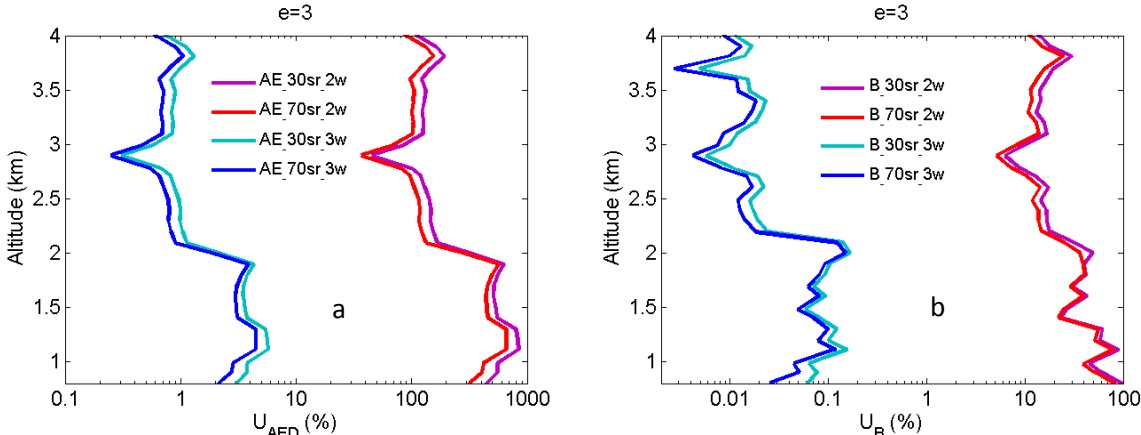


Fig.10 NO$_2$ number density relative uncertainty owing to aerosol extinction (a) and backscatter (b) with
e=3.

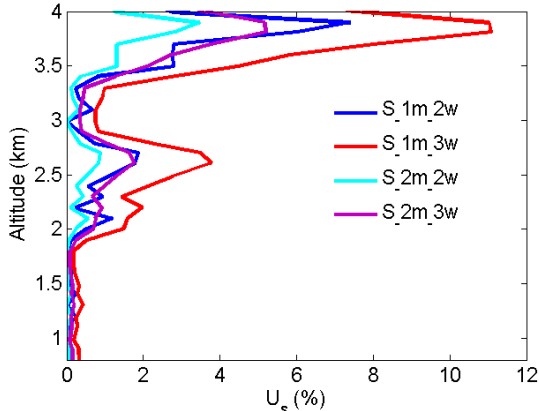


Fig.11 NO$_2$ number density relative uncertainty owing to the noise of signals with average of 1 minute
and 2 minutes.

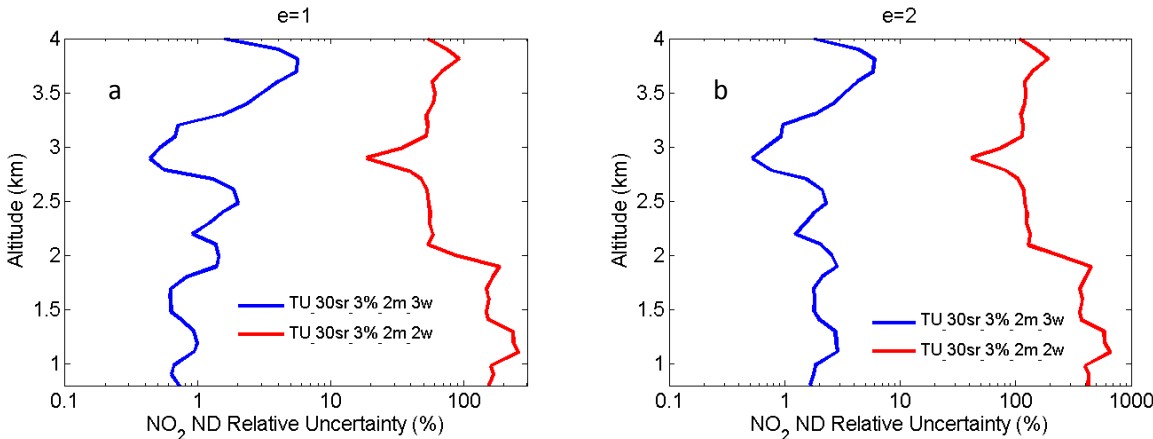



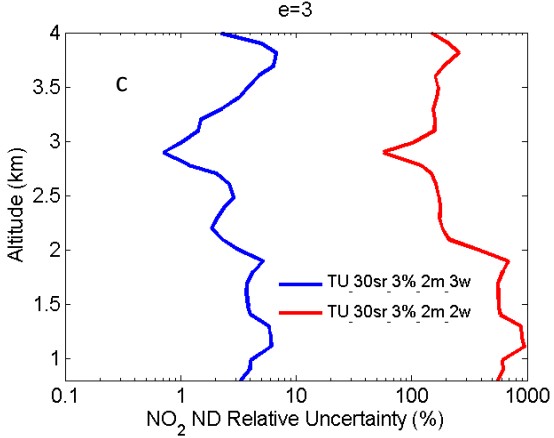


Fig.12 $NO_2$ number density total relative uncertainty with e=1 (a), e=2 (b) and e=3 (c).

**5. Results**
The three-wavelength DIAL technique was implemented by the HU lidar measurements during
two cases at night and the resulting vertical profiles are presented in Fig. 13. All $NO_2$ lidar
measurements presented here are obtained at times with less than 10% cloud coverage below 8
km. HU lidar 438 nm (blue line), 439.5 nm (red line) and 441 nm (black line) elastic signals
measured at 21:00 (local time) on May 13, 2020 and 22:00 (local time) on July 27, 2020 are
shown in Fig. 13 (a) and (c), respectively. The average integration time for these signals is 2
minutes. Determined from the lidar elastic signals in Fig. 13 (a) and (c), there is an existing
aerosol layer between 2.2 km and 3.5 km on May 13, while July 27 presented a clean atmosphere.
Fig. 13 (b) and (d) show retrieved $NO_2$ profiles using the three-wavelength DIAL technique (red
line). The black error bars in Fig.13 (b) and (d) indicate the uncertainty of $NO_2$ retrieval
calculated using Eq. (13). In Fig. 13 (b), the retrieved $NO_2$ profile between 2.2 km and 3.5 km on
May 13 is smooth and not affected by the aerosol layer. The $NO_2$ profiles (sky-blue line and
purple line) were also retrieved using the conventional two-wavelength DIAL technique without
and with aerosol correction shown in Fig. 13 (b) resulting in a bump between 2.2 km and 3.5 km
in the $NO_2$ profile retrieved using the two-wavelength DIAL technique. This inconsistency
suggests that the two-wavelength DIAL technique cannot remove AED of the aerosol layer
between 2.2 km and 3.5 km and the retrieved $NO_2$ profile contains AED interference. Moreover,
the $NO_2$ retrievals below 2 km using two-wavelength DIAL technique shown in Fig. 13 (b) and
(d) are more than the three-wavelength DIAL technique suggesting that the AED of boundary
aerosol was not correctly removed. Aerosol correction is very important for $NO_2$ retrieval using
the conventional two-wavelength DIAL technique [Sasano et al., 1985]. These results suggest
that the proposed three-wavelength DIAL technique can effectively remove influence of aerosol
on the retrieval of $NO_2$. As a first-order assessment of the HU lidar $NO_2$ profiles, we compare the
retrieval results to simulated data from the Weather Research and Forecasting Chemistry (WRF-
Chem) model (Grell et al., 2005) at 12 km × 12 km spatial resolution and 200 m vertical
resolution. Past studies have demonstrated that WRF-Chem simulated $NO_2$ results show good
agreement between the OMI satellite measurements and aircraft measurements [Amnuaylojaroen
et al., 2019; Barten et al., 2020] providing a data source to examine the accuracy of the HU
retrievals using both two-wavelength DIAL technique and three-wavelength DIAL technique.
The HU local $NO_2$ profiles for these two cases are simulated using WRF-Chem model and
shown in Fig. 13 (b) and (d). WRF-Chem simulated $NO_2$ magnitudes tend to be lower compared
to HU retrieved $NO_2$ profiles using three-wavelength DIAL technique (typically within ±0.1
ppb), except above 3.5 km on May 13, 2020, however, the comparison demonstrates a consistent
vertical profile shape between observations and the model simulation. And retrieval results using
the three-wavelength DIAL technique are much closer to simulated values compared to using the
two-wavelength DIAL technique. These figures also demonstrate that the reduced fluctuations
caused by aerosol backscatter when using the three-wavelength DIAL technique results in
vertical profiles of $NO_2$ which are much more consistent with simulated data when compared to
results of the two-wavelength DIAL retrievals. Both the WRF-Chem simulated profiles and the
HU retrievals of NO$_2$ using three-wavelength DIAL technique are associated with uncertainties
which could result in the differences in magnitude; however, given the consistent nature in the
vertical profile shapes from both data sources provides confidence that the HU lidar is retrieving
NO$_2$ vertical profiles using three-wavelength DIAL technique in the troposphere.

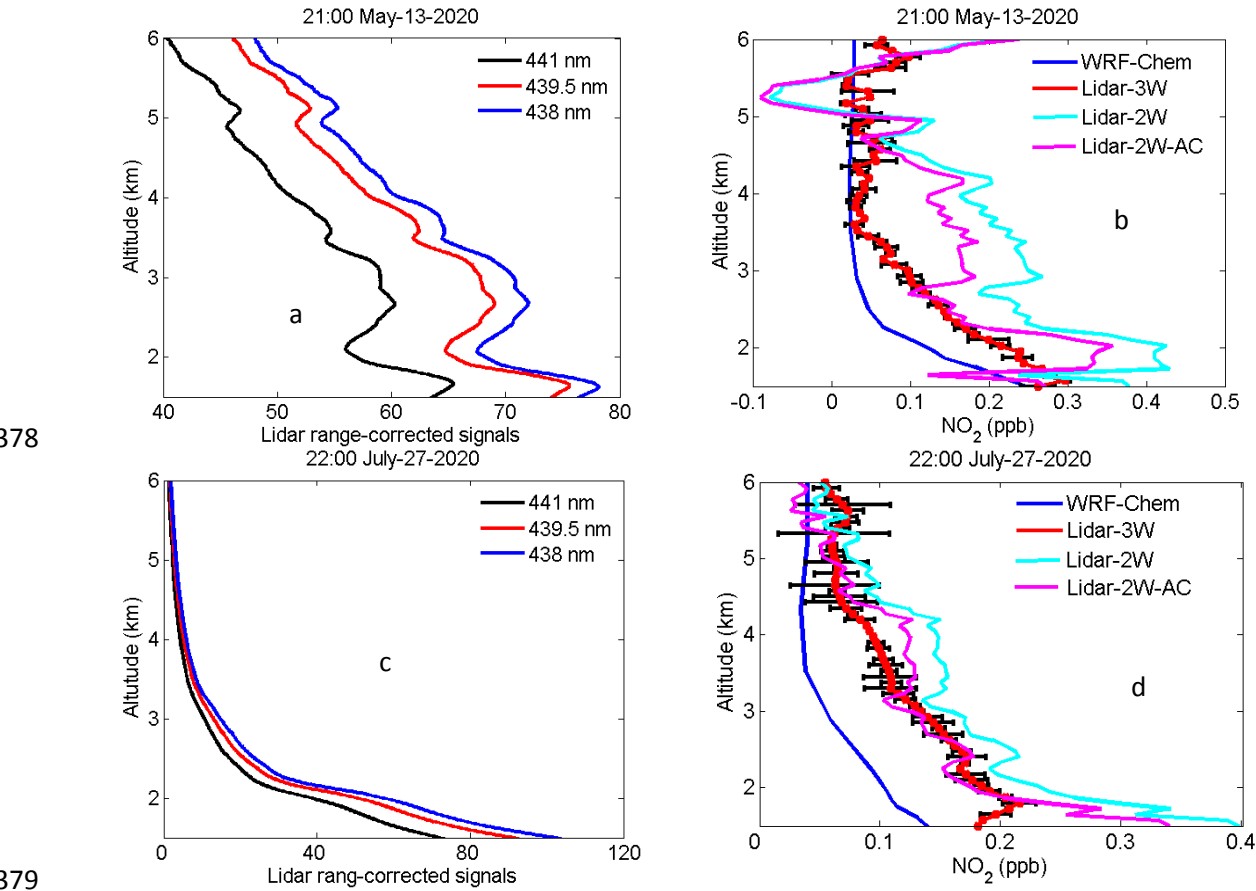



Fig.13 HU lidar 438 nm, 439.5 nm and 441 nm elastic signals measured at 21:00 (local time) on May 13,
2020 (a) and 22:00 (local time) on July 27, 2020 (c); NO$_2$ profiles obtained using three-wavelength DIAL
technique, two-wavelength DIAL technique and WRF-Chem model at 21:00 on May 13, 2020 (b) and
22:00 on July 27, 2020 (d).

**6. Conclusion**
This study describes a lidar retrieval technique using three wavelengths simultaneously emitted
from an OPO laser to measure tropospheric NO$_2$ profiles. The three-wavelength DIAL retrieval
equations describe how the retrievals decrease errors caused by aerosol interference. Aerosol
extinction differences using this proposed technique can be decreased to less than 2% of aerosol
extinction differences resulting from a conventional two-wavelength DIAL technique.
Comparing the HU lidar results to WRF-Chem model output demonstrates that the $NO_2$
magnitudes and vertical structure are in much better agreement with simulated data when
applying the three-wavelength DIAL technique compared to using the two-wavelength technique.
In the future, we will add new filters to obtain daytime $NO_2$ measurements. We also plan to
purchase $NO_2$ balloonsondes for acquiring true validation data to evaluate HU lidar $NO_2$ results.

**Code availability**
The software code for this paper is available from the first author.

**Data availability**
The HU lidar data are archived at http://cas.hamptonu.edu/data-products/. DISCOVER-AQ data
and WRF-Chem model output data are available from the first authors upon request.

**Author contributions**
JS designed the three-wavelength OPO DIAL system, did simulation works for the system,
developed the $NO_2$ retrieval algorithm, and prepared the original manuscript. PMM was
responsible for funding acquisition. MSJ provided simulated data from WRF-Chem model to
verify $NO_2$ retrieval in this study. JTS, MJN, TAB, SK and GPG contributed to the analysis of
$NO_2$ retrieval uncertainty. All listed authors contributed to the review and editing of this paper.

**Competing interests**
The authors declare that they have no conflict of interest.

**Acknowledgments**
We thank NASA TOLNet and NASA DISCOVER-AQ measurements for our simulation work.
Matthew Johnson's contribution was supported by the NASA's TOLNet Science Team and the
Tropospheric Composition Program. We also thank Dr. Gabriele Pfister from the Atmospheric
Chemistry Observations & Modeling Lab at the National Center for Atmospheric Research for
providing the WRF-Chem calculation applied in this study.
**Financial support**
This study was supported by the PIRT project funded by US Army Research, Development and
Engineering Command (AQC) Center (DOD) under HU PIRT Award # 551150-211150) and the
National Oceanic and Atmospheric Administration- Cooperative Science Center for Earth
System and Sciences and Remote Sensing Technologies (NOAA-CESSRST) under the
Cooperative Agreement Grant #: NA16SEC4810008. The statements contained within the
manuscript/research article are not the opinions of the funding agency or the U.S. government,
but reflect the author's opinions.
**Review statement**
This paper was reviewed by Dr. Piet Stammes (Editor) and one anonymous referee.
**Supplementary Material**
S.1 Two-wavelength DIAL retrieval equation

Table s.1. Extinction and backscatter of molecule and aerosol for wavelengths of $\lambda_1$ $and$ $\lambda_2$.

| wavelength | Molecular backscattering | Aerosol backscattering | Molecular extinction | Aerosol extinction |
|---|---|---|---|---|
| $\lambda_1$ | $\left(\dfrac{\lambda_1}{\lambda_2}\right)^{-4} \beta_m$ | $\left(\dfrac{\lambda_1}{\lambda_2}\right)^{-e} \beta_a$ | $\left(\dfrac{\lambda_1}{\lambda_2}\right)^{-4} \alpha_m$ | $\left(\dfrac{\lambda_1}{\lambda_2}\right)^{-e} \alpha_a$ |
| $\lambda_2$ | $\beta_m$ | $\beta_a$ | $\alpha_m$ | $\alpha_a$ |

The two elastic lidar equations can be expressed as:

$$X(\lambda_1, Z) = C_1 \frac{\left[\left(\frac{\lambda_1}{\lambda_2}\right)^{-4}\beta_m(Z) + \left(\frac{\lambda_2}{\lambda_1}\right)^{-e}\beta_a(Z)\right]}{Z^2} \exp\left\{-2\int_0^Z \left[\left(\frac{\lambda_1}{\lambda_2}\right)^{-4}\alpha_m(z) + \left(\frac{\lambda_2}{\lambda_1}\right)^{-e}\alpha_a(z) + \sigma_N(\lambda_1,z)N_N(z) + O_{abs}(\lambda_1,z)\right]dz\right\} \quad \text{(s.1)}$$

$$X(\lambda_2, Z) = C_2 \frac{[\beta_m(Z) + \beta_a(Z)]}{Z^2} \exp\left\{-2\int_0^Z [\alpha_m(z) + \alpha_a(z) + \sigma_N(\lambda_2,z)N_N(z) + O_{abs}(\lambda_2,z)]dz\right\} \quad \text{(s.2)}$$

where $X$ is the lidar signal; $C_1$ and $C_2$ are lidar constants; the subscripts $a$ and $m$ represent aerosol,
and molecule, respectively; $\sigma_N$ is the absorption cross section for the gas of interest; $N_N$ is the
molecular density of the gas of interest; $O_{abs}$ is absorption of gases other than the gas of interest
and $z$ is the altitude. The molecular density of the gas of interest can be obtained after taking
ratio of Eq. (1) to Eq. (2).
NO$_2$ density retrieval equation can be expressed as Eq. (3):
$$N_N(Z) = \frac{\frac{1}{2} \times \frac{d}{dz}\left[\ln\frac{X(\lambda_1,Z)}{X(\lambda_2,Z)}\right] - AED(z) - MED(z) - OAD(z) - B(z)}{\Delta\sigma_N} \quad \text{(s.3)}$$
$$\Delta\sigma_N = \sigma_N(\lambda_2) - \sigma_N(\lambda_1) \quad \text{(s.4)}$$
$$B(z) = \frac{1}{2}\frac{d}{dz}\left[\ln\frac{\left(\frac{\lambda_1}{\lambda_2}\right)^{-4}\beta_m(z) + \left(\frac{\lambda_1}{\lambda_2}\right)^{-e}\beta_a(z)}{\beta_m(z) + \beta_a(z)}\right] \quad \text{(s.5)}$$
$$AED(z) = \left[1 - \left(\frac{\lambda_1}{\lambda_2}\right)^{-e}\right]\alpha_a(Z) \quad \text{(s.6)}$$
$$MED(z) = \left[1 - \left(\frac{\lambda_1}{\lambda_2}\right)^{-4}\right]\alpha_m(Z) \quad \text{(s.7)}$$
$$OAD(z) = O_{abs}(\lambda_2, z) - O_{abs}(\lambda_1, z) \quad \text{(s.8)}$$
S.2 Units for all variables
Table s.2. Units for all variables

| Items | Description | Unit |
|---|---|---|
| C1, C2,C3 | lidar constant | constant (number) |
| $\alpha$ | extinction coefficient | km$^{-1}$ |
| $\beta$ | backscattering coefficient | km$^{-1}$sr$^{-1}$ |
| $\lambda$ | wavelength | nm |
| $\sigma$ | absorption cross section | cm$^2$molecule$^{-1}$ |
| Z, z | altitude | km |
| X | lidar range-corrected signal | mv |
| N | number density | molecule/cm$^3$ |

| S | lidar ratio | sr$^{-1}$ |
|---|---|---|
| U | Relative uncertainty | % |

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
