# Peer review of "Optical Parametric Oscillator Differential Absorption Lidar"

_Atmospheric Measurement Techniques, 2020_

## Referee Comment (RC1)

Review for J. Su et al., Tropospheric NO2 Measurements Using a Three-wavelength Optical Parametric Oscillator Differential Absorption Lidar submitted to Atmospheric Measurement Techniques.

This paper provides a preliminary analysis of a three-wavelength (dual-DIAL) lidar technique applied to both a modeled lidar signal and to lidar signals from the OPO-based Hampton University $NO_2$ lidar. While multi-wavelength DIAL methods do exist, the wavelength flexibility of new laser transmitters permit novel optimizations to improve lidar measurements of chemical species in the atmosphere. The authors demonstrate that many of the errors in the $NO_2$ lidar profiles (due to uncertainties in the analysis parameters) are significantly reduced using this method, and they compare the lidar results to results from a well-established atmospheric composition model.

I believe that this paper would benefit from revision to provide more details/discussion of a few essential points and for several technical corrections.

In section 2 it would be useful to cite a few examples of prior works on three-wavelength (or dual-DIAL) lidar analysis. (There are many available, and previous work is suggested in lines 135ff, but that citation is missing/mislabeled in the references section.)

In equation 5, it would be helpful to label this term as NAD since it is described as this in the text. (e.g. "NAD $= \Delta\sigma_N = \dots$")

In equation 7, it would be useful to include the term K that will be used later (e.g. "AED $= \dots = K\alpha_a(Z)$")

Since the three-wavelength results are compared against the two-wavelength results throughout this paper, it would be useful to readers to include the two-wavelength equations that correspond to equations 4 through 9.

In section 2a, it would be useful to note that DIAL systems for other atmospheric gases like ozone, it is only practical to use wavelength selection Method B because of the shape of the ozone absorption spectrum (lacking narrow peaks). The shape of the absorption spectrum of $NO_2$ allows for an especially favorable three-wavelength analysis using Method C because it is possible to choose the points spanning over the peak as shown in figures 1 and 2.

In sections 2 and 3, it is mentioned that the wavelengths are optimized according to the rules a. (maximize NAD) and b. (minimize AED), but this is a multivariate optimization. It would be useful to provide more detail of the optimization process and how the authors arrived at the final wavelengths.

There are two sections of text (lines 163ff and lines 223ff) that describe the lidar hardware and should be combined.

In section 4, Disregarding the uncertainty introduced by the lidar signals, $U_s$, should not be taken lightly. In particular, because this term is the result of taking a derivative of (logs of ratios of) signals, it can be very susceptible to noise in the raw signals. The modelled analysis in this paper uses relatively noise-free aerosol and ozone profiles (figure 4) which reduces this issue and facilitates evaluation of the optimizations presented. It is noted that the lidar signals in section 4 were integrated for 2 minutes to reduce the signal noise, and as a result, the resulting $NO_2$ profiles are relatively smooth. However, there should be some discussion of this noise source and its contribution to the resulting $NO_2$ profile uncertainty. This provides readers with an estimation of the relative contribution of signal noise which ultimately depends on lidar specifications (e.g. power and aperture) as well as on temporal and spatial resolution (i.e. averaging).

In section 4, the notation of "$N_a$" and "$\sigma_a$" for number density of air and Rayleigh (air) scattering cross section, respectively, might be less confusing as "$N_m$" and "$\sigma_m$" to be consistent with the rest of the text where "m" denotes molecular terms and "a" denotes aerosol terms.

In section 4 (line 272), a reference to Fernald's paper describing the lidar inversion procedure should be provided.

In section 5 (line326), the vertical resolution of the WRF-Chem results should be provided since the comparison with the lidar will be in this dimension.

Typographic errors:
Line 118 (equation 4), the numerator of the lidar signal term should be $X(\lambda_1,Z)X(\lambda_3,Z)$.
Lines 193-194, "…light blue lines are NAD." should be "…light blue lines are OAD."
Line 217 (figure 5 caption), Add "B" to the list of values described as shown on the graphs.
Line 228 "NO2" should be "$NO_2$"
Line 239, "NO" should be "$NO_2$"
Line 257 (equation 18), "$In$" should be "$ln$"
Lines 271ff, Font used for "$\alpha_a$" is different than that used elsewhere (e.g. compare with line 109).
Line 309, "Fig. 8" should be "Fig. 12"

General review/editing for grammar would be helpful.

---

## Author Comment (AC1)

Dear Reviewer,

We have revised our manuscript based on your comments. We thank you very much for your insightful comments which we have used to greatly improve our manuscript. Below, please find our specific responses (in blue) along with each of your comments. A track changes version of the manuscript shows all changes marked with red.

Thank you so much for taking care of our submission.

Sincerely,

Jia Su

Comments to Authors:

1. In section 2 it would be useful to cite a few examples of prior works on three-wavelength (or dual-DIAL) lidar analysis. (There are many available, and previous work is suggested in lines 135ff, but that citation is missing/mislabeled in the references section.)

**We added a few examples of prior works on three-wavelength lidar analysis from line 137 to line 141 and the missing reference at line 463.**

"Wang used three wavelengths corresponding to the strong, medium and weak absorption of $O_3$ to obtain an accurate stratospheric ozone profile in the presence of volcanic aerosols. Liu used three wavelengths of 448.10nm, 447.20nm and 446.60 nm corresponding to the strong, medium and weak absorption of $NO_2$ to retrieve $NO_2$."

2. In equation 5, it would be helpful to label this term as NAD since it is described as this in the text. (e.g. "NAD = $\Delta\sigma$N = …")

**The $\Delta\sigma$N is the absorption cross section difference for $NO_2$. In general, $\sigma$ is expressed as cross section, so we think $\Delta\sigma$N is better than the expression of NAD.**

3. In equation 7, it would be useful to include the term K that will be used later (e.g. "AED = … = K$\alpha$a(Z)")

**We revised it at line 121 according to reviewer's comments.**

4. Since the three-wavelength results are compared against the two-wavelength results throughout this paper, it would be useful to readers to include the two-wavelength equations that correspond to equations 4 through 9.

**We added the two-wavelength equations as supplements.**

5. In section 2a, it would be useful to note that DIAL systems for other atmospheric gases like ozone, it is only practical to use wavelength selection Method B because of the shape of the ozone absorption spectrum (lacking narrow peaks). The shape of the absorption spectrum of NO2 allows for an especially favorable three-wavelength analysis using Method C because it is possible to choose the points spanning over the peak as shown in figures 1 and 2.

**We added it from line 147 to line 149 according to reviewer's comments.**

"However, for DIAL systems to measure other atmospheric gases like ozone, it is only practical

to use wavelength selection Method B because of the shape of the ozone absorption spectrum

(lacking narrow peaks)."

6. In sections 2 and 3, it is mentioned that the wavelengths are optimized according to the rules a. (maximize NAD) and b. (minimize AED), but this is a multivariate optimization. It would be useful to provide more detail of the optimization process and how the authors arrived at the final wavelengths.

**Our two rules are increasing absorption cross section difference of $NO_2$ and decreasing AED. From Eq.12, AED can be determined by the value of K. We can choose the appropriate three wavelengths to make the value of K equal or close to 0, the value of AED will be equal or close to 0. We added explanation for it from line 179 to line 180. We simulated B using three-wavelength Dial technique and two-wavelength Dial technique, and found that three-wavelength Dial technique can decrease the value of B.**

"Combining the OPO laser energy outputs, $NO_2$ absorption spectral and two three-wavelength chosen rules, 438 nm, 439.5 nm and 441 nm shown in Fig. 2 result in the wavelengths of HU three-wavelength DIAL system because $\Delta\sigma_N$ of the three-wavelength pair is more than other three-wavelength pairs in $NO_2$ strong absorption spectral zone and the $K$ value of the three-wavelength is close to 0."

7. There are two sections of text (lines 163ff and lines 223ff) that describe the lidar hardware and should be combined.

**We revised it at line 229 according to reviewer's comments.**

8. In section 4, Disregarding the uncertainty introduced by the lidar signals, Us, should not be taken lightly. In particular, because this term is the result of taking a derivative of (logs of ratios of) signals, it can be very susceptible to noise in the raw signals. The modelled analysis in this paper uses relatively noise-free aerosol and ozone profiles (figure 4) which reduces this issue and facilitates evaluation of the optimizations presented. It is noted that the lidar signals in section 4 were integrated for 2 minutes to reduce the signal noise, and as a result, the resulting NO2 profiles are relatively smooth. However, there should be some discussion of this noise source and its contribution to the resulting NO2 profile uncertainty. This provides readers with an estimation of the relative contribution of signal noise which ultimately depends on lidar specifications (e.g. power and aperture) as well as on temporal and spatial resolution (i.e. averaging).

**We added an equation (line 263) and analysis (between line 287 and line 295) for uncertainties owing to noise of three-wavelength lidar signals.**

$$U_S(z) = \frac{u\left\{\frac{1}{2}\times\frac{d}{dz}\left[ln\frac{X(\lambda_1,Z)(\lambda_3,Z)}{X(\lambda_2,Z)^2}\right]\right\}}{N_N(Z)\Delta\sigma_N} =$$

$$\frac{\frac{1}{2}\times\sqrt{\left(\frac{d\left\{\frac{d}{dz}\left[ln\frac{X(\lambda_1,Z)(\lambda_3,Z)}{X(\lambda_2,Z)^2}\right]\right\}}{d[X(\lambda_1,Z)]}\times u[X(\lambda_1,Z)]\right)^2 + \left(\frac{d\left\{\frac{d}{dz}\left[ln\frac{X(\lambda_1,Z)(\lambda_3,Z)}{X(\lambda_2,Z)^2}\right]\right\}}{d[X(\lambda_2,Z)]}\times u[X(\lambda_2,Z)]\right)^2 + \left(\frac{d\left\{\frac{d}{dz}\left[ln\frac{X(\lambda_1,Z)(\lambda_3,Z)}{X(\lambda_2,Z)^2}\right]\right\}}{d[X(\lambda_3,Z)]}\times u[X(\lambda_3,Z)]\right)^2}}{N_N(Z)\Delta\sigma_N}$$

$$(18)$$

From Eq. (18), $U_s$ is determined by uncertainties of three-wavelength lidar signals. The uncertainties of lidar signals with average integration time of 1 minute and 2 minutes are derived from Poisson statistics associated with the probability of detection of a repeated random event [Measures, 1984; Leblanc et al., 2016]. NO$_2$ number density relative uncertainty owing to the noise of lidar signals with average integration time of 1 minute and 2 minutes are obtained shown in Fig11. We can see $U_s$ using two-wavelength DIAL technique is smaller than using three-wavelength DIAL technique. With increase of average integration time from 1 minute to 2 minutes, $U_s$ can be effectively reduced.

[Figure]

Fig.11 NO$_2$ number density relative uncertainty owing to the noise of signals with average of 1 minute and 2 minutes.

9. In section 4, the notation of "Na" and "σa" for number density of air and Rayleigh (air) scattering cross section, respectively, might be less confusing as "Nm" and "σm" to be consistent with the rest of the text where "m" denotes molecular terms and "a" denotes aerosol terms.

**We revised them from line 254 to line 256 according to reviewer's comments.**

10. In section 4 (line 272), a reference to Fernald's paper describing the lidar inversion procedure should be provided.

**We added a reference to Fernald's paper at line 430 according to reviewer's comments.**

"Fernald, F. G., Herman, B. M., and Reagan, J. A.: Determination of aerosol height distributions by lidar, Journal of Applied meteorology., 11(3), 482-489, 1972"

10. In section 5 (line326), the vertical resolution of the WRF-Chem results should be provided since the comparison with the lidar will be in this dimension.

**We added it at line 343 according to reviewer's comments.**

"As a first-order assessment of the HU lidar NO$_2$ profiles, we compare the retrieval results to simulated data from the Weather Research and Forecasting Chemistry (WRF-Chem) model (Grell et al., 2005) at 12 km × 12 km spatial resolution and 200 m vertical resolution. "

11. Typographic errors:

Line 118 (equation 4), the numerator of the lidar signal term should be X(λ1,Z)X(λ3,Z).

**We revised it at line 118.**

Lines 193-194, "…light blue lines are NAD." should be "…light blue lines are OAD."

**We revised it at line 200.**

Line 217 (figure 5 caption), Add "B" to the list of values described as shown on the graphs.

**We added it at line 223.**

Line 228 "NO2" should be "NO$_2$"

**We revised it at line 234.**

Line 239, "NO" should be "NO$_2$"

**We revised it at line 245.**

Line 257 (equation 18), "In" should be "ln"

**We revised it at line 261.**

Lines 271ff, Font used for "αa" is different than that used elsewhere (e.g. compare with line 109).

**We revised it from line 265 to line 270.**

Line 309, "Fig. 8" should be "Fig. 12"

**We revised it at line 326.**

---

## Author Comment (AC2)

Dear Reviewer,

We have revised our manuscript based on your comments. We thank you very much for your insightful comments which we have used to greatly improve our manuscript. Below, please find our specific responses (in blue) along with each of your comments. Thank you so much for taking care of our submission.

Sincerely,

Jia Su

Comments to Authors:

1. 65-67: This statement on satellite observations is not correct. With the recent high spatial resolution observations of NO2 by TROPOMI on Sentinel-5P since 2017, with 3.5 x 5.5. km2 pixels, plumes of NO2 by cities, power plants, and even ships can be tracked. Appropriate references to the novel TROPOMI NO2 observations should be given, for example:

- Lorente, A., Boersma, K.F., Eskes, H.J. et al., Quantification of nitrogen oxides emissions from build-up of pollution over Paris with TROPOMI. Sci Rep 9, 20033 (2019). https://doi.org/10.1038/s41598-019-56428-5

- Georgoulias et al., Detection of NO2 pollution plumes from individual ships with the TROPOMI/S5P satellite sensor, Environ. Res. Lett. 15, 124037, 2020

**We modified our statement on satellite observations according to reviewer's comments and added above two references for the manuscript.**

2. 74: a reference to an NO2 sonde system should be given: Sluis, W. W., Allaart, M. A. F., Piters, A. J. M., and Gast, L. F. L.: The development of a nitrogen dioxide sonde, Atmos. Meas. Tech., 3, 1753–1762, https://doi.org/10.5194/amt-3-1753-2010, 2010.

**We added the reference for the manuscript.**

3. Section 2:

Here consistency is needed, and proper introduction of formulae.

- l. 79: 24x7 operation
- l. 85: absorption > absorbing

- l. 87: quant

- l. 82 ff: subscripts that are words, like "on" and "off", should be in upright font.

- Please give the units of all quantities used: \lambda, \beta, \alpha, etc. etc.

**We modified them according to reviewer's comments.**

- Eqs. 1-3:
  - Are these relations only valid for an upright pointing lidar? **NO, they are valid for all lidars.**
  - Both capital Z and small z are used as variables. Are they the same height variable?? This consistency question holds throughout the paper. **Both capital Z and z are height variable. The capital Z is the height of lidar signal, while z is height variable of integral formula.**
  - Please give the units of X, C1, C2, C3, \sigma, N, etc. **We added them in our manuscript.**

l. 124 and other places: acronyms like AED should be in upright font; only symbols are in slant font. **We modified them according to reviewer's comments.**

- l. 116: derivatives w.r.t. which variable? **We modified it. "The molecular density of the gas of interest can be obtained using Eqs. (1), (2) and (3)."**

- What does the D mean in AED, MED, OAD ? **D means difference.**
  - l. 129: unclear sentence. The text of the method description should be clarified. **We modified it. "For correction of AED and B, we need accurate aerosol measurements. However, accurate aerosol measurements are not easily to be obtained. From the above NO2 retrieval relative equation, AED are determined by the three wavelengths, so how to choose the three wavelengths is very critical to reduce AED and improve the NO2 retrievals accuracy."**

4. In comparison to Eqs. 4-9, how does the two-wavelength DIAL NO2 retrieval equation look like? This is relevant for the comparisons shown later on. **We added the two-wavelength equations as supplements.**

5. The description of the A, B, C methods in Sect. 2 should be improved:

- l. 135-136: These methods A and B have not been introduced yet. Please give a name for the methods: increasing absorption, decreasing absorption, and maximum in absorption **We used suggested increasing absorption, decreasing absorption, and maximum absorption as names for methods A, B and C.**

- l. 137: missing reference. Or should it be Liang? **We added the missing reference.**

- l. 142-143: please compare to the two-wavelength DIAL equation. Please explain why eq. 12 is better. **We added explanation for it.**

- Eq. 12: is the +-sign the most important difference between the three methods? **We added them according to reviewer's comments.**

- l. 152: please first introduce the derivation of Eq. 13. **We revised it.**

- l. 153: what does K represent? what is the relation to the earlier equations? **K is part of Eq.(7) which is related to three wavelengths.**

- l. 156: please give some examples for the three wavelengths driven by K. **We added some examples in manuscript.**

- Eq. 13: please first show the equation, then discuss it. **We revised it according to reviewer's comments.**

6. Sect. 3:

- l. 169: what about the wavelengths below 400 nm? **NO2 have strong absorption between 420 nm and 450 nm, so we selected the wavelengths above 400 nm.**

- l. 172: what about the relative weight of the two rules/criteria?

- l. 177: how does the two-wavelength NO2 retrieval equation looks like? **We added the two-wavelength equations as supplements.**

- l. 182: please give the physical unit of the lidar ratio. **The lidar ratio is the ratio of aerosol extinction coefficients to aerosol backscattering coefficients.**

- l. 190: which HITRAN version? **The version is 1.1.2.0**

- l. 191: e in italics **We revised it.**

- l. 193-194: this line colour code information belongs in the figure caption. **We revised it.**

- l. 200: far less > much smaller **We used "much smaller" instead of "far less". .**

- l. 223: This is a strange order of this section: first instrument description, then simulation, and then again instrument part. Please restructure section 3 into two subsections: (1) instrument description, (2) simulation of the retrieval. **We adjusted order for the section in our manuscript.**

- l. 239: missing words at the end of this sentence ? **Yes, we revised it.**

(7) Sect. 4:

This section on error analysis requires drastic improvement: clarification, better introduction of equations, consistency with the rest of the paper, correction of grammar and typo's.

- l. 242: … from standard uncertainty: please explain.**, Leblanc introduced standard uncertainties for Lidar (Eq.(2) in his reference)**

- Eq. 14: what is the unit of U? **The unit of U is "%".**

- l. 246: and not discussed in this work: unclear. **We added relative uncertainties for Lidar signal noise.**

- l. 249: what is the uncertainty function u? please give reference. **u are standard uncertainties. Capital U are relative standard uncertainties. And we added a reference for it.**

- l. 249: how are these equations 15-18 derived? please first show the equations, and then explain the variables in them. **We revised it.**

- l. 250: in Section 2 the subscript 'a' means aerosols, and 'm' means molecules. It is very confusing that here 'a' means air. Please be consistent. **We used "m" instead of "a".**

- l. 260 ff: 8-hour or eight-hour: be consistent **We revised it.**

- l. 272: Fernald's method: give reference. **We added a reference for Fernald's method.**

8. Sect. 5:

- What is vertical resolution of the measured NO2 profiles? **The vertical resolution of measured NO2 profiles is 100 m.**

- l. 327-328: … good agreement between the OMI ... > with the OMI satellite measurements **We revised it.**

- Fig. 12: what is the reason of the still large differences between measurement and model ?

 **Matthew S. Johnson (co-author) did a lot work about comparing NO2 profiles from aircraft measurements and WRF-Chem model, and said NO2 profiles from WRF-Chem model are generally a little lower than real NO2 profiles.**

**Figures**

- Captions should be self-explanatory! **We revised it.**

- Fig. 2: unclear alignment of the wavelengths in the legend. What is the source of this NO2 absorption cross-section spectrum? Caption: …. strong absorption cross-section spectrum ...

**We downloaded absorption data of NO2 from Hitran and plotted it using Matlab.**

**We revised alignment of the wavelength in the legend.**

- Fig. 3: Are this figure and table taken from the manufacturer's brochure? It should be a new figure for this manuscript, otherwise there is a copyright issue. **Yes. Thanks! We redo them.**

- Fig. 5: Please indicate in the legends whether it is 2- or 3-wavelength DIAL. Explain the quantities in the caption. What is meant with the x-axis label difference of ... ? between what and what? **We revised it according to reviewer's comments.**

- Please combine figures 8, 9 and 10. Only e is varying. Explain the legends in the caption. **We revised it according to reviewer's comments.**

- Fig. 11, caption: 'except $U\_s$': what does this mean?? explain what e is. what is TU? explain the legends. **We revised it according to reviewer's comments.**

- Fig. 12: explain the black error bars in b and d. **We added an explanation for them.**

**Textual**

There are several typos and grammatical mistakes. Please carefully check the English language throughout the manuscript.

Often, the article is missing, e.g. on l. 36: … from the WRF-Chem model….

The singular/plural should be checked, e.g. l. 42: … the main emission sources …

Check typography of the references.

**We revised it according to reviewer's comments.**

---

## Author Response (AR1)

Dear Editor and Reviewer,

We have revised our manuscript based on your comments. We thank you very much for your insightful comments which we have used to greatly improve our manuscript. Below, please find our specific responses (in blue) along with each of your comments. A track changes version of the manuscript shows all changes marked with red.

Thank you so much for taking care of our submission.

Sincerely,

Jia Su

**Editor (Dr. Piet Stammes):**

Comments to Authors:

1. 65-67: This statement on satellite observations is not correct. With the recent high spatial resolution observations of NO2 by TROPOMI on Sentinel-5P since 2017, with 3.5 x 5.5. km2 pixels, plumes of NO2 by cities, power plants, and even ships can be tracked. Appropriate references to the novel TROPOMI NO2 observations should be given, for example:

- Lorente, A., Boersma, K.F., Eskes, H.J. et al., Quantification of nitrogen oxides emissions from build-up of pollution over Paris with TROPOMI. Sci Rep 9, 20033 (2019). https://doi.org/10.1038/s41598-019-56428-5

- Georgoulias et al., Detection of NO2 pollution plumes from individual ships with the TROPOMI/S5P satellite sensor, Environ. Res. Lett. 15, 124037, 2020

**We modified our statement on satellite observations according to reviewer's comments from line 65 to line 70 and added above two references for the manuscript.**

"Moreover, plumes of $NO_2$ by cities, power plants, and even ships can be tracked using the recent high spatial resolution observations of $NO_2$ from TROPOMI on Sentinel-5P since 2017 [Lorente, et al., 2019; Georgoulias et al., 2020]. However, they are unable to obtain local high temporal resolution $NO_2$ emissions such as variations in hourly $NO_2$ concentrations due to their long repeat cycle, since the lifetime of tropospheric $NO_2$ is only about 6 hour in summer and 18-24 hours in winter due to photochemical effect [Beirle, et al., 2003; Cui et al., 2016]."

2. 74: a reference to an NO2 sonde system should be given: Sluis, W. W., Allaart, M. A. F., Piters, A. J. M., and Gast, L. F. L.: The development of a nitrogen dioxide sonde, Atmos. Meas. Tech., 3, 1753–1762, https://doi.org/10.5194/amt-3-1753-2010, 2010.

**We added the reference for the manuscript.**

3. Section 2:

Here consistency is needed, and proper introduction of formulae.

- l. 79: 24x7 operation
- l. 85: absorption > absorbing
- l. 87: quant
- l. 82 ff: subscripts that are words, like "on" and "off", should be in upright font.
- Please give the units of all quantities used: \lambda, \beta, \alpha, etc. etc.

**We modified them according to reviewer's comments.**

- Eqs. 1-3:
  - Are these relations only valid for an upright pointing lidar?

    **NO, they are valid for all lidars.**

  - Both capital Z and small z are used as variables. Are they the same height variable?? This consistency question holds throughout the paper.

    **Both capital Z and z are height variable. The capital Z is the height of lidar signal, while z is height variable of integral formula.**

  - Please give the units of X, C1, C2, C3, \sigma, N, etc.

    **We added them in our manuscript as supplements.**

l. 124 and other places: acronyms like AED should be in upright font; only symbols are in slant font.

**We modified them according to reviewer's comments.**

  - l. 116: derivatives w.r.t. which variable?

    **We modified it at line 118.**

- What does the D mean in AED, MED, OAD ?

  **D means difference.**

  - l. 129: unclear sentence. The text of the method description should be clarified.

**We modified it from line 131 to line 135.**

4. In comparison to Eqs. 4-9, how does the two-wavelength DIAL NO2 retrieval equation look like? This is relevant for the comparisons shown later on.

**We added the two-wavelength equations as supplements.**

5. The description of the A, B, C methods in Sect. 2 should be improved:

- l. 135-136: These methods A and B have not been introduced yet. Please give a name for the methods: increasing absorption, decreasing absorption, and maximum in absorption

**We used suggested increasing absorption, decreasing absorption, and bumping absorption as names for methods A, B and C.**

- l. 137: missing reference. Or should it be Liang?

**We added the missing reference.**

- l. 142-143: please compare to the two-wavelength DIAL equation. Please explain why eq. 12 is better.

**The $\Delta\sigma_N$ for two-wavelength DIAL is $\Delta\sigma_N = \sigma_N(\lambda_2) - \sigma_N(\lambda_1)$.**

**The $\Delta\sigma_N$ for three-wavelength DIAL is**

$$\Delta\sigma_N = abs[\sigma_N(\lambda_2) - \sigma_N(\lambda_1)] + abs[\sigma_N(\lambda_2) - \sigma_N(\lambda_3)]$$

**$\Delta\sigma_N$ for three-wavelength DIAL is sum of absolute value of $abs[\sigma_N(\lambda_2) - \sigma_N(\lambda_1)]$ and $abs[\sigma_N(\lambda_2) - \sigma_N(\lambda_3)]$. So $\Delta\sigma_N$ for three-wavelength DIAL is better.**

- Eq. 12: is the +-sign the most important difference between the three methods?

**Yes, + means NO2 absorption cross section difference is increase.**

- l. 152: please first introduce the derivation of Eq. 13.

**We revised it.**

- l. 153: what does K represent? what is the relation to the earlier equations?

**K is a part of Eq.(7) which is related to three wavelengths.**

- Eq. 13: please first show the equation, then discuss it.

**We revised it according to reviewer's comments.**

6. Sect. 3:

- l. 169: what about the wavelengths below 400 nm?

**NO$_2$ have strong absorption between 420 nm and 450 nm, so we selected the wavelengths above 400 nm.**

- l. 172: what about the relative weight of the two rules/criteria?

**The rule to increase differences of the NO$_2$ absorption cross section is prior to the rule to reduce or remove AED. The relative weights for them are 55% and 45%.**

- l. 177: how does the two-wavelength NO2 retrieval equation looks like?

**We added the two-wavelength equations as supplements.**

- l. 182: please give the physical unit of the lidar ratio.

**The lidar ratio is the ratio of aerosol extinction coefficients to aerosol backscattering coefficients. The unit is sr$^{-1}$.**

- l. 190: which HITRAN version?

**The version is 1.1.2.0**

- l. 191: e in italics

**We revised it.**

- l. 193-194: this line colour code information belongs in the figure caption. **We added it to Fig.5 caption from line 249 to line 252.**

- l. 200: far less > much smaller

**We used "much smaller" instead of "far less" at line 234.**

- l. 223: This is a strange order of this section: first instrument description, then simulation, and then again instrument part. Please restructure section 3 into two subsections: (1) instrument description, (2) simulation of the retrieval.

**We adjusted order for the section in our manuscript.**

- l. 239: missing words at the end of this sentence ?

**Yes, we revised it at line 208.**

(7) Sect. 4:

This section on error analysis requires drastic improvement: clarification, better introduction of equations, consistency with the rest of the paper, correction of grammar and typo's.

- l. 242: … from standard uncertainty: please explain.**,**

**Leblanc introduced standard uncertainties for DIAL lidar retrieval (Eq.(4) in his reference). The relative uncertainty = standard uncertainty /measured value × 100%.**

- Eq. 14: what is the unit of U?

**The unit of U is "%".**

- l. 246: and not discussed in this work: unclear.

**We added relative uncertainties for Lidar signal noise.**

- l. 249: what is the uncertainty function u? please give reference.

**u are standard uncertainties. Capital U are relative uncertainties. And we added a reference for it.**

- l. 249: how are these equations 15-18 derived? please first show the equations, and then explain the variables in them.

**These equations are derived using lidar uncertainty propagation formula.**

- l. 250: in Section 2 the subscript 'a' means aerosols, and 'm' means molecules. It is very confusing that here 'a' means air. Please be consistent.

**We used "m" instead of "a" at line 278.**

- l. 260 ff: 8-hour or eight-hour: be consistent

**We revised it from line 283 to line 284.**

- l. 272: Fernald's method: give reference.

**We added a reference for Fernald's method.**

8. Sect. 5:

- What is vertical resolution of the measured NO2 profiles?

**The average vertical resolution for measured $NO_2$ profiles is 200 m.**

- l. 327-328: … good agreement between the OMI ... > with the OMI satellite measurements

**We revised it.**

- Fig. 12: what is the reason of the still large differences between measurement and model ?

**Matthew S. Johnson (co-author) did a lot of work about comparing NO2 profiles from aircraft measurements and WRF-Chem model, and said NO2 profiles from WRF-Chem model are generally a little lower than real NO2 profiles.**

- Captions should be self-explanatory!

**We revised them.**

- Fig. 2: unclear alignment of the wavelengths in the legend. What is the source of this NO2 absorption cross-section spectrum? Caption: …. strong absorption cross-section spectrum ...

**We downloaded absorption data of NO2 from Hitran and plotted it using Matlab.**

**We revised alignment of the wavelength in the legend of Fig.2.**

- Fig. 3: Are this figure and table taken from the manufacturer's brochure? It should be a new figure for this manuscript, otherwise there is a copyright issue.

**Yes. Thanks! We redo them.**

- Fig. 5: Please indicate in the legends whether it is 2- or 3-wavelength DIAL. Explain the quantities in the caption. What is meant with the x-axis label difference of ... ? between what and what?

**We revised Fig.6 according to reviewer's comments. For 3-wavelength DIAL, the difference means the difference of 438 nm, 439.5 nm and 441 nm. For 2-wavelength DIAL, the difference means the difference of 438 nm and 439.5 nm.**

- Please combine figures 8, 9 and 10. Only e is varying. Explain the legends in the caption.

**We think there are too many lines in one figure and readers cannot see these lines clearly if we combined figure 8, 9 and 10.**

- Fig. 11, caption: 'except U_s': what does this mean?? explain what e is. what is TU? explain the legends.

**We revised it according to reviewer's comments. e is angstrom exponent for aerosol. Please see line Line 263 and Eq. 13 for TU.**

- Fig. 12: explain the black error bars in b and d.

**We added an explanation for them at line 344.**

Textual

There are several typos and grammatical mistakes. Please carefully check the English language throughout the manuscript.

Often, the article is missing, e.g. on l. 36: … from the WRF-Chem model….

The singular/plural should be checked, e.g. l. 42: … the main emission sources …

Check typography of the references.

**We revised it according to reviewer's comments.**

**Reviewer:**

Comments to Authors:

1. In section 2 it would be useful to cite a few examples of prior works on three-wavelength (or dual-DIAL) lidar analysis. (There are many available, and previous work is suggested in lines 135ff, but that citation is missing/mislabeled in the references section.)

**We added a few examples of prior works on three-wavelength lidar analysis from line 141 to line 146 and the missing reference.**

2. In equation 5, it would be helpful to label this term as NAD since it is described as this in the text. (e.g. "NAD = ΔσN = …")

**The ΔσN is the absorption cross section difference for NO₂. In general, σ is expressed as cross section, so we think ΔσN is better than the expression of NAD.**

3. In equation 7, it would be useful to include the term K that will be used later (e.g. "AED = … = Kαa(Z)")

**We revised it at line 124 according to reviewer's comments.**

4. Since the three-wavelength results are compared against the two-wavelength results throughout this paper, it would be useful to readers to include the two-wavelength equations that correspond to equations 4 through 9.

**We added the two-wavelength DIAL equations as supplements.**

5. In section 2a, it would be useful to note that DIAL systems for other atmospheric gases like ozone, it is only practical to use wavelength selection Method B because of the shape of the ozone absorption spectrum (lacking narrow peaks). The shape of the absorption spectrum of NO2 allows for an especially favorable three-wavelength analysis using Method C because it is possible to choose the points spanning over the peak as shown in figures 1 and 2.

**We added it from line 154 to line 156 according to reviewer's comments.**

6. In sections 2 and 3, it is mentioned that the wavelengths are optimized according to the rules a. (maximize NAD) and b. (minimize AED), but this is a multivariate optimization. It would be useful to provide more detail of the optimization process and how the authors arrived at the final wavelengths.

**Our two rules are increasing absorption cross section difference of NO$_2$ and decreasing AED. From Eq.12, AED can be determined by the value of K. We can choose the appropriate three wavelengths to make the value of K equal or close to 0, the value of AED will be equal or close to 0. We added explanation for it from line 179 to line 180. We simulated B using three-wavelength Dial technique and two-wavelength Dial technique, and found that three-wavelength Dial technique can decrease the value of B. We added the optimization process from line 190 to line 192.**

7. There are two sections of text (lines 163ff and lines 223ff) that describe the lidar hardware and should be combined.

**We revised it according to reviewer's comments.**

8. In section 4, Disregarding the uncertainty introduced by the lidar signals, Us, should not be taken lightly. In particular, because this term is the result of taking a derivative of (logs of ratios of) signals, it can be very susceptible to noise in the raw signals. The modelled analysis in this paper uses relatively noise-free aerosol and ozone profiles (figure 4) which reduces this issue and facilitates evaluation of the optimizations presented. It is noted that the lidar signals in section 4 were integrated for 2 minutes to reduce the signal noise, and as a result, the resulting NO2 profiles are relatively smooth. However, there should be some discussion of this noise source and its contribution to the resulting NO2 profile uncertainty. This provides readers with an estimation

of the relative contribution of signal noise which ultimately depends on lidar specifications (e.g. power and aperture) as well as on temporal and spatial resolution (i.e. averaging).

**We added an equation (line 271) and analysis (between line 302 and line 310) for uncertainties owing to noise of three-wavelength lidar signals.**

9. In section 4, the notation of "Na" and "σa" for number density of air and Rayleigh (air) scattering cross section, respectively, might be less confusing as "Nm" and "σm" to be consistent with the rest of the text where "m" denotes molecular terms and "a" denotes aerosol terms.

**We revised them from line 278 to line 286 according to reviewer's comments.**

10. In section 4 (line 272), a reference to Fernald's paper describing the lidar inversion procedure should be provided.

**We added a reference to Fernald's paper according to reviewer's comments.**

10. In section 5 (line326), the vertical resolution of the WRF-Chem results should be provided since the comparison with the lidar will be in this dimension.

**We added it at line 359 according to reviewer's comments.**

11. Typographic errors:

Line 118 (equation 4), the numerator of the lidar signal term should be $X(\lambda_1,Z)X(\lambda_3,Z)$.

Lines 193-194, "…light blue lines are NAD." should be "…light blue lines are OAD."

Line 217 (figure 5 caption), Add "B" to the list of values described as shown on the graphs.**.**

Line 228 "NO2" should be "NO2"

Line 239, "NO" should be "NO2"

Line 257 (equation 18), "In" should be "ln"

Lines 271ff, Font used for "αa" is different than that used elsewhere (e.g. compare with line 109).

Line 309, "Fig. 8" should be "Fig. 12"

**We revised them according to reviewer's comments. Thanks!**